# MeGraph: Capturing Long-Range Interactions by Alternating Local and Hierarchical Aggregation on Multi-Scaled Graph Hierarchy

**Honghua Dong**[1,2,3*†]   **Jiawei Xu**[1,4*]   **Yu Yang**[1,4*]   **Rui Zhao**[1]   **Shiwen Wu**[5]
**Chun Yuan**[4]   **Xiu Li**[4]   **Chris Maddison**[2,3]   **Lei Han**[1†]
[1]Tencent Robotics X   [2]University of Toronto   [3]Vector Institute
[4]Tsinghua University   [5]Hong Kong University of Science and Technology

## Abstract

Graph neural networks, which typically exchange information between local neighbors, often struggle to capture long-range interactions (LRIs) within the graph. Building a graph hierarchy via graph pooling methods is a promising approach to address this challenge; however, hierarchical information propagation cannot entirely take over the role of local information aggregation. To balance locality and hierarchy, we integrate the local and hierarchical structures, represented by intra- and inter-graphs respectively, of a multi-scale graph hierarchy into a single mega graph. Our proposed `MeGraph` model consists of multiple layers alternating between local and hierarchical information aggregation on the mega graph. Each layer first performs local-aware message-passing on graphs of varied scales via the intra-graph edges, then fuses information across the entire hierarchy along the bidirectional pathways formed by inter-graph edges. By repeating this fusion process, local and hierarchical information could intertwine and complement each other. To evaluate our model, we establish a new Graph Theory Benchmark designed to assess LRI capture ability, in which `MeGraph` demonstrates dominant performance. Furthermore, `MeGraph` exhibits superior or equivalent performance to state-of-the-art models on the Long Range Graph Benchmark. The experimental results on commonly adopted real-world datasets further demonstrate the broad applicability of `MeGraph`. [1]

## 1   Introduction

Graph-structured data, such as social networks, traffic networks, and biological data, are prevalent across a plethora of real-world applications. Recently, Graph Neural Networks (GNNs) have emerged as a powerful tool for modeling and understanding the intricate relationships and patterns present in such data. Most existing GNNs learn graph representations by iteratively aggregating information from individual nodes' local neighborhoods through the message-passing mechanism. Despite their effectiveness, these GNNs struggle to capture long-range interactions (LRIs) between nodes in the graph. For instance, when employing a 4-layer vanilla GNN on the 9-node (*A* to *I*) graph (as shown in Fig. 1), the receptive field of node *A* is limited to 4-hop neighbors, making the aggregation of information from nodes *G*, *H*, and *I* into node *A* quite challenging. While GNNs could theoretically incorporate information from nodes $n$-hops away with $n$-layers of message passing, this often leads to over-smoothing and over-squashing issues [17, 3] when $n$ is large.

---

[*]Equal Contribution. Work done while HD, JX and YY are interns at Tencent Robotics X.

[†]Corresponding authors, contact `honghuad@cs.toronto.edu` and `lxhan@tencent.com`.

[1]Project website and open-source code can be found at `https://sites.google.com/view/megraph`.

37th Conference on Neural Information Processing Systems (NeurIPS 2023).

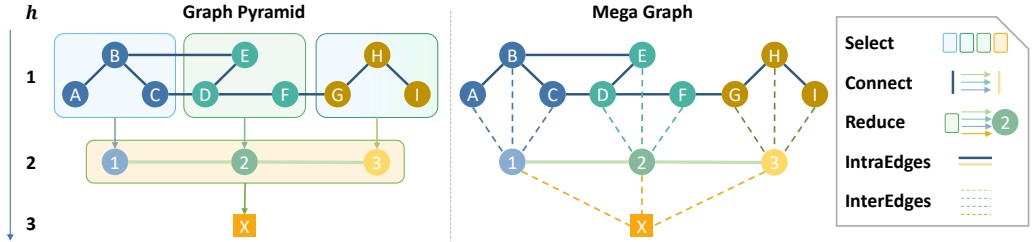

Figure 1: Illustration of the graph pooling operation, graph pyramid, and mega graph. **Graph pooling** is a downsampling process comprising SELECT, CONNECT, and REDUCE steps. It begins by selecting subsets for grouping and each subset collapses into a new node in the pooled graph. Next, it forms new edges by merging the original ones, and finally calculates the pooled graph's features. In this graph, nodes *A B C*, *D E F*, and *G H I* are pooled into nodes *1*, *2*, and *3* respectively, while the edges (*B*, *E*) and (*C*, *D*) are merged into (*1*, *2*). **Graph Pyramid** involves multi-scaled graphs derived from iterative graph pooling, with the height indicating different scales and $h = 1$ symbolizing the original graph. **Mega Graph** is formed by connecting the graph pyramid using inter-graph edges, which are the by-products of graph pooling.

One mainstream solution to this problem involves constructing a multi-scale graph hierarchy through graph pooling methods. Previous efforts, such as Graph UNets [22] and HGNet [47], have attempted to broaden the receptive field using this strategy. They downsample and upsample the graph, aggregating information along the hierarchy. However, hierarchical information propagation cannot take over the role of local information aggregation. To illustrate, consider the graph hierarchy depicted in Fig. 1. The information propagated along the hierarchy from node *B* to nodes *D*, *E*, and *F* tends to be similar since they share the common path *B-1-X-2*. However, in the original graph, node *B* holds different degrees of importance to nodes *D*, *E*, and *F* as they are 2, 1, and 3 hops away respectively.

To balance the importance of locality and hierarchy, we amalgamate the local and hierarchical structures of a multi-scale graph hierarchy into a single mega graph as depicted in Fig. 1, where we refer to the local structure as intra-graph edges and hierarchical structure as inter-graph edges. Based on this mega graph, we introduce our MeGraph model consisting of $n$ Mee layers. Each layer first performs local-aware message-passing on graphs of varied scales via the intra-graph edges and then fuses information across the whole hierarchy along the bidirectional pathways formed by inter-graph edges. This method enables hierarchically fused information to circulate within local structures and allows locally fused information to distribute across the hierarchy. By repeating this fusion process, local and hierarchical information could intertwine and complement each other. Moreover, to support flexible graph pooling ratios when constructing the multi-scale graph hierarchy, we propose a new graph pooling method S-EdgePool that improves from EdgePool [13].

In our experiments, We first evaluate MeGraph's capability to capture Long Range interactions (LRIs). We establish a Graph Theory Benchmark comprising four tasks related to shortest paths and one related to connected components. MeGraph demonstrates superior performance compared with many competitive baselines. MeGraph also achieves comparable or superior performance than the state-of-the-art on the Long Range Graph Benchmark (LRGB) [17]. In addition, we perform extensive experiments on widely-used real-world datasets that are not explicitly tailored for assessing the capacity to capture LRIs. These include the GNN benchmark [15] and OGB-G datasets [28]. In these tests, MeGraph demonstrates superior or equivalent performance compared to the baseline models, suggesting its broad applicability and effectiveness.

The main contributions of this work are summarized as follows: 1) **Mega graph and novel architecture**: we propose the mega graph, a multi-scale graph formed by intra- and inter-graph edges, where the message-passing over the mega graph naturally balances locality and hierarchy. On this basis, we introduce a novel architecture MeGraph, which alternates information aggregation along the intra- and inter-edges of the mega graph. This fusion process intertwines local and hierarchical information, leading to mutual benefits. 2) **Hierarchical information fusion**: we design a bidirectional pathway to facilitate information fusion among the hierarchies. 3) **S-EdgePool**: we enhance EdgePool into S-EdgePool, allowing an adjustable pooling ratio. 4) **Benchmark and Evaluations**: we establish a new graph theory benchmark to evaluate the ability of models to capture LRIs. In these evaluations, MeGraph exhibits dominant performance. Additionally, MeGraph achieves new SOTA in one task of LRGB and shows better or comparable performance compared with baselines on popular real-world datasets.

## 2 Notations, Backgrounds and Preliminaries

Let $\mathcal{G} = (\mathcal{V}, \mathcal{E})$ be a graph with node set $\mathcal{V}$ (of cardinality $N^v$) and edge set $\mathcal{E}$ (of cardinality $N^e$). The edge set can be represented as $\mathcal{E} = \{(s_k, t_k)\}_{k=1:N^e}$, where $s_k$ and $t_k$ are the indices of the source and target nodes connected by edge $k$. We define $\mathbf{X}^{\mathcal{G}}$ as features of graph $\mathcal{G}$, which is a combination of global (graph-level) features $\mathbf{u}^{\mathcal{G}}$, node features $\mathbf{V}^{\mathcal{G}}$, and edge features $\mathbf{E}^{\mathcal{G}}$. Accordingly, we use $\mathbf{V}_i^{\mathcal{G}}$ to represent the features of a specific node $v_i$, and $\mathbf{E}_k^{\mathcal{G}}$ denotes the features of a specific edge $(s_k, t_k)$. We may abuse the notations by omitting the superscript $\mathcal{G}$ when there is no context ambiguity.

### 2.1 Graph Network (GN) Block

We adopt the Graph Network (GN) block design in accordance with the GN framework [6]. In our notation, a GN block accepts a graph $\mathcal{G}$ and features $\mathbf{X} = (\mathbf{u}, \mathbf{V}, \mathbf{E})$ as inputs, and produces new features $\mathbf{X}' = (\mathbf{u}', \mathbf{V}', \mathbf{E}')$. A full GN block [6] includes the following update steps. In each of these steps, $\phi$ denotes an update function, typically implemented as a neural network:

**Edge features**: $\mathbf{E}_k' = \phi^e(\mathbf{E}_k, \mathbf{V}_{s_k}, \mathbf{V}_{t_k}, \mathbf{u}), \forall k \in [1, N^e]$.
**Node features**: $\mathbf{V}_i' = \phi^v(\rho^{e \to v}(\{\mathbf{E}_k'\}_{k \in [1,N^e], t_k=i}), \mathbf{V}_i, \mathbf{u}), \forall i \in [1, N^v]$, where $\rho^{e \to v}$ is an aggregation function taking the features of incoming edges as inputs.
**Global features**: $\mathbf{u}' = \phi^u(\rho^{e \to u}(\mathbf{E}'), \rho^{v \to u}(\mathbf{V}'), \mathbf{u})$, where $\rho^{e \to u}$ and $\rho^{v \to u}$ are two global aggregation functions over edge and node features.

Given a fixed graph structure $\mathcal{G}$ and the consistent input and output formats outlined above, GN blocks can be seamlessly integrated to construct complex, deep graph networks.

### 2.2 Graph Pooling

Graph pooling operation downsamples the graph structure and its associated features while ensuring the preservation of structural and semantic information inherent to the graph. Drawing from the SRC framework [23], we identify graph pooling as a category of functions, POOL, that maps a graph $\mathcal{G} = (\mathcal{V}, \mathcal{E})$ with $N^v$ nodes and features $\mathbf{X}^{\mathcal{G}}$ to a reduced graph $\tilde{\mathcal{G}} = (\tilde{\mathcal{V}}, \tilde{\mathcal{E}})$ with $N^{\tilde{v}}$ nodes and new features $\mathbf{X}^{\tilde{\mathcal{G}}}$. Here, $N^{\tilde{v}} \leq N^v$ and $(\tilde{\mathcal{G}}, \mathbf{X}^{\tilde{\mathcal{G}}}) = \text{POOL}(\mathcal{G}, \mathbf{X}^{\mathcal{G}})$.

The SRC framework deconstructs the POOL operation into SELECT, REDUCE, and CONNECT functions, which encompass most existing graph pooling techniques. We reinterpret these functions in our own notation as follows:

$$(\hat{\mathcal{G}}, \mathbf{X}^{\hat{\mathcal{G}}}) = \text{SELECT}(\mathcal{G}, \mathbf{X}^{\mathcal{G}}); \quad \tilde{\mathcal{G}} = \text{CONNECT}(\mathcal{G}, \hat{\mathcal{G}}, \mathbf{X}^{\hat{\mathcal{G}}}); \quad \mathbf{X}^{\tilde{\mathcal{G}}} = \text{REDUCE}(\mathbf{X}^{\mathcal{G}}, \hat{\mathcal{G}}, \mathbf{X}^{\hat{\mathcal{G}}}). \quad (1)$$

As shown in Fig. 1, the SELECT establishes $N^{\tilde{v}}$ nodes for the pooled graph, and each node $\tilde{v}$ corresponds to a subset of nodes $S_{\tilde{v}} \subseteq \mathcal{V}$ in the input graph. This creates an *undirected* bipartite graph $\hat{\mathcal{G}} = (\hat{\mathcal{V}}, \hat{\mathcal{E}})$, with $\hat{\mathcal{V}} = \mathcal{V} \cup \tilde{\mathcal{V}}$ and $(v, \tilde{v}) \in \hat{\mathcal{E}}$ if and only if $v \in S_{\tilde{v}}$. We refer to this graph $\hat{\mathcal{G}}$ as the *inter-graph*, a larger graph that links nodes in the input graph $\mathcal{G}$ with nodes in the pooled graph $\tilde{\mathcal{G}}$. The SELECT function can be generalized to include inter-graph features $\mathbf{X}^{\hat{\mathcal{G}}}$. As an example, edge weights can be introduced for some edge $(\hat{s}_k, \hat{t}_k)$ in graph $\hat{\mathcal{G}}$ to gauge the importance of node $\hat{s}_k$ from the input graph contributing to node $\hat{t}_k$ in the pooled graph.

The CONNECT function reconstructs the edge set $\tilde{\mathcal{E}}$ between the nodes in $\tilde{\mathcal{V}}$ of the pooled graph $\tilde{\mathcal{G}}$ based on the original edges in $\mathcal{E}$ and the inter-graph edges in $\hat{\mathcal{E}}$. The REDUCE function calculates the graph features $\mathbf{X}^{\tilde{\mathcal{G}}}$ of graph $\tilde{\mathcal{G}}$ by aggregating input graph features $\mathbf{X}^{\mathcal{G}}$, taking into account both the inter-graph $\hat{\mathcal{G}}$ and features $\mathbf{X}^{\hat{\mathcal{G}}}$. In a similar vein to the relationship between graph lifting and coarsening, we define the EXPAND function for graph features, which serves as the inverse of the REDUCE function: $\mathbf{X}^{\mathcal{G}} = \text{EXPAND}(\mathbf{X}^{\tilde{\mathcal{G}}}, \hat{\mathcal{G}}, \mathbf{X}^{\hat{\mathcal{G}}})$.

## 3 Methods

We begin with the introduction of the mega graph (Sec.3.1), which amalgamates the local (intra-edges) and hierarchical (inter-edges) structures of a multi-scale graph hierarchy into a single graph.

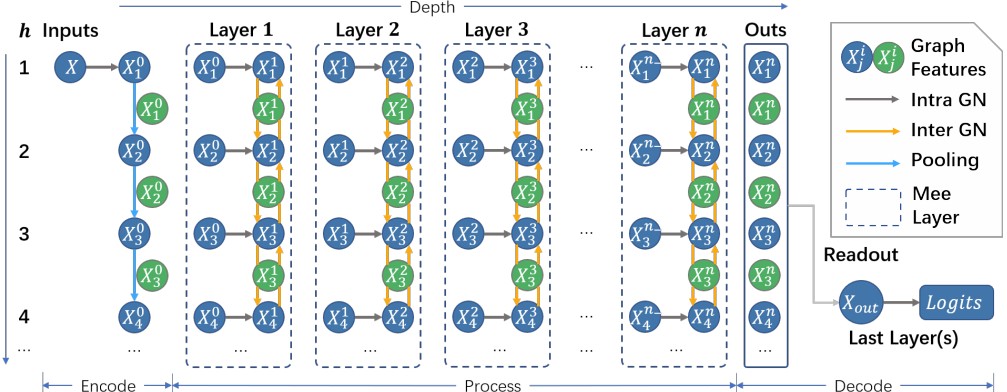

Figure 2: Illustration of the `MeGraph` model, where $n_-$ denotes $n-1$. The blue and green circles represent features of intra- and inter-graphs, respectively. In this figure, the horizontal and vertical directions represent the interaction among the local structure (intra-graph) and graph hierarchy (inter-graph) respectively. The features of intra- and inter-graphs are represented by blue and green circles, respectively. In this figure, the horizontal and vertical directions denote the local structure and graph hierarchy respectively. During the *encode* stage, the mega graph is constructed using graph pooling. In the *process* stage, the `Mee` layer, which features bidirectional pathways across multiple scales, is stacked $n$ times. In the *decode* stage, multi-scale features are read out. The golden inter GN blocks form bidirectional pathways across the whole hierarchy.

Following this, we present the `MeGraph` model (Sec.3.2), which alternates between the aggregation of local and hierarchical information along the intra- and inter-edges of the mega graph. We then discuss the specific choices made for the core modules of the `MeGraph`, along with the innovations (Sec.3.3). Finally, we delve into the computational complexity of the `MeGraph` model (Sec.3.4).

## 3.1 Connecting Multi-scale Graphs into a Mega Graph

Similar to the concept of an image pyramid [1], a graph pyramid is constructed by stacking multi-scale graphs, which are obtained through iterative downsampling of the graph using a graph pooling technique. Formally, in alignment with the definition of an image feature pyramid [39], we define a graph feature pyramid as a set of graphs $\boldsymbol{\mathcal{G}}_{1:h} := \{\mathcal{G}_i\}_{i=1,\cdots,h}$ and their corresponding features $\mathbf{X}^{\boldsymbol{\mathcal{G}}_{1:h}} := \{\mathbf{X}^{\mathcal{G}_i}\}_{i=1,\cdots,h}$. Here, $\mathcal{G}_1$ represents the original graph, $\mathbf{X}^{\mathcal{G}_1}$ signifies the initial features, $h$ stands for the *height*, and $(\mathcal{G}_i, \mathbf{X}^{\mathcal{G}_i}) = \texttt{POOL}(\mathcal{G}_{i-1}, \mathbf{X}^{\mathcal{G}_{i-1}})$ for $i > 1$.

By iteratively applying the `POOL` function, we can collect the inter-graphs $\hat{\boldsymbol{\mathcal{G}}}_{1:h} := \{\hat{\mathcal{G}}_i\}_{i=1,\cdots,h-1}$ and their features $\mathbf{X}^{\hat{\boldsymbol{\mathcal{G}}}_{1:h}} := \{\mathbf{X}^{\hat{\mathcal{G}}_i}\}_{i=1,\cdots,h-1}$ (since there are $h-1$ inter-graphs for $h$ intra-graphs), where $(\hat{\mathcal{G}}_i, \mathbf{X}^{\hat{\mathcal{G}}_i}) = \texttt{SELECT}(\mathcal{G}_i, \mathbf{X}^{\mathcal{G}_i})$ for $i < h$. The bipartite inter-graph $\hat{\mathcal{G}}$ and its features $\mathbf{X}^{\hat{\mathcal{G}}}$ essentially depict the relationships between the graphs before and after the pooling process (see Sec. 2.2).

Finally, as illustrated in Fig. 1, we wire the graph pyramid $\boldsymbol{\mathcal{G}}_{1:h}$ using the edges found in the bipartite graphs $\hat{\boldsymbol{\mathcal{G}}}_{1:h}$. This results in a mega graph $\mathcal{MG} = (\mathcal{MV}, \mathcal{ME})$, where $\mathcal{MV} = \bigcup_{i=1}^{h} \mathcal{V}_i$ and $\mathcal{ME} = \bigcup_{i=1}^{h} \mathcal{E}_i \cup \bigcup_{i=1}^{h-1} \hat{\mathcal{E}}_i$. The structure of the mega graph would vary as the graph pooling method trains. We denote $\mathcal{MG}_{\text{intra}} = \bigcup_{i=1}^{h} \mathcal{G}_i$ as the intra-graph of $\mathcal{MG}$, and refer to the edges therein as intra-edges. Correspondingly, $\mathcal{MG}_{\text{inter}} = \bigcup_{i=1}^{h-1} \hat{\mathcal{G}}_i$ is referred to as the inter-graph of $\mathcal{MG}$, with its corresponding edges termed as inter-edges. The features $\mathbf{X}^{\mathcal{MG}}$ of the mega graph $\mathcal{MG}$ is a combination of intra-graph features $\mathbf{X}^{\boldsymbol{\mathcal{G}}_{1:h}}$ and inter-graph features $\mathbf{X}^{\hat{\boldsymbol{\mathcal{G}}}_{1:h}}$.

## 3.2 Mega Graph Message Passing

We introduce the `MeGraph` architecture, designed to perform local and hierarchical aggregations over the mega graph alternately. As shown in Fig.2, the architecture follows the *encode-process-decode* design [6, 25] and incorporates GN blocks (refer to Sec. 2.1) as fundamental building blocks.

During the *encode* stage, initial features are inputted into an intra-graph GN block, which is followed by a sequence of graph pooling operations to construct the mega graph $\mathcal{MG}$ and its associated features

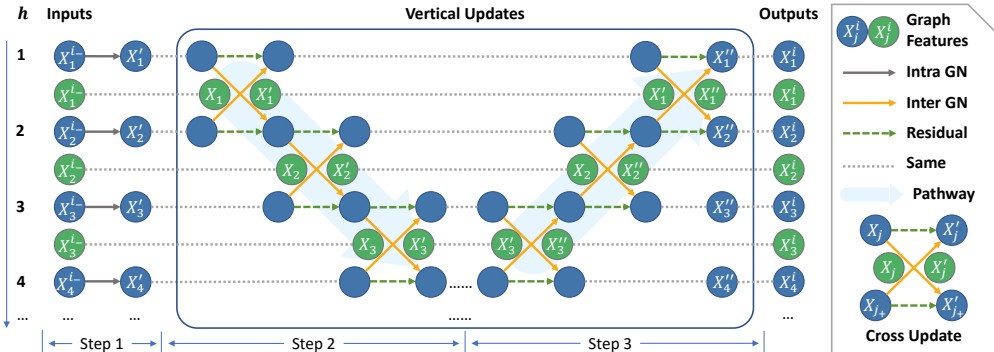

Figure 3: Illustration of the `Mee` layer, where $i_-$ denotes $i-1$ and $j_+$ denotes for $j+1$. The blue and green circles represent the features of intra- and inter-graphs, respectively. Grey and golden arrows represent the intra and inter GN blocks. The cross-update utilizes inter GN blocks to exchange information between consecutive heights, as elaborated in the main text. The `Mee` layer first aggregates information locally along inter-graph edges. It then applies cross-updates sequentially from lower to higher levels, accumulating information along the pathway to pass to the higher hierarchy. The process is reversed in the last step.

$(\mathbf{X}^0)^{\mathcal{MG}}$. In the *process* stage, the `Mee` layer, which performs both local and hierarchical information aggregation within the mega graph, is stacked $n$ times. The $i$-th `Mee` layer receives $(\mathbf{X}^{i-1})^{\mathcal{MG}}$ as input and outputs $(\mathbf{X}^i)^{\mathcal{MG}}$. Through the stacking of `Mee` layers, a deeper architecture is created, enabling a more profound fusion of local and hierarchical information. Lastly, in the *decode* stage, the features $(\mathbf{X}^n)^{\mathcal{MG}}$ are transformed into task-specific representations using readout functions.

**Mee Layer.** The `Mee` layer is designed to aggregate local and hierarchical information within the mega graph. A detailed structure of the `Mee` layer is depicted in Fig. 3.

For the $i$-th `Mee` layer, we consider inputs denoted by $(\mathbf{X}^{i-1})^{\mathcal{MG}} = \{(\mathbf{X}^{i-1})^{\mathcal{G}_{1:h}}, (\mathbf{X}^{i-1})^{\hat{\mathcal{G}}_{1:h}}\}$. For simplicity, we omit the superscript and denote the features of intra- and inter-graphs as $\{\mathbf{X}_j^{i-1}\}_{j=1,\cdots,h} := (\mathbf{X}^{i-1})^{\mathcal{G}_{1:h}}$ and $\{\hat{\mathbf{X}}_j^{i-1}\}_{j=1,\cdots,h-1} := (\mathbf{X}^{i-1})^{\hat{\mathcal{G}}_{1:h}}$ respectively.

The first step applies GN blocks on intra-graph edges, performing message passing on the local structure of graphs at each scale: $\mathbf{X}_j' = \mathrm{GN}_{\mathrm{intra}}^{i,j}(\mathcal{G}_j, \mathbf{X}_j^{i-1})$. Here, $\mathbf{X}_j'$ represents the updated intra-graph $\mathcal{G}_j$ features.

The second and third steps focus on multi-scale information fusion. The second step applies cross-updates across consecutive heights from $1$ to $h$, while the third step reverses the process, forming a bidirectional pathway for the information flow across the hierarchy. The cross-update between consecutive heights $j$ and $j+1$ is denoted by a function $(\mathbf{X}_j', \hat{\mathbf{X}}_j', \mathbf{X}{j+1}') = \texttt{X-UPD}(j, \mathbf{X}_j, \hat{\mathbf{X}}_j, \mathbf{X}{j+1})$. The prime notation indicates the updated value, and residual links [26] are used in practice.

This cross-update can be implemented via an inter-graph convolution with $\mathrm{GN}_{\mathrm{inter}}^{i,j}$, referred to as *X-Conv* (detailed in App.C.1). Alternatively, it can be realized using the `REDUCE` and `EXPAND` operations of `POOL` (refer to Sec.2.2) by $\mathbf{X}_{j+1}' = \texttt{REDUCE}(\hat{\mathcal{G}}_j, \hat{\mathbf{X}}_j^0, \mathbf{X}_j)$ and $\mathbf{X}_j' = \texttt{EXPAND}(\hat{\mathcal{G}}_j, \hat{\mathbf{X}}_j^0, \mathbf{X}_{j+1})$, where $\hat{\mathcal{G}}_j$ is the $j$-th inter-graph. We denote this implementation as *X-Pool*.

The `Mee` layer outputs features $\{\mathbf{X}_j^i\}_{j=1,\cdots,h}$ and $\{\hat{\mathbf{X}}_j^i\}_{j=1,\cdots,h-1}$. Residual links [26] can be added from $\mathbf{X}_j^{i-1}$ to $\mathbf{X}_j^i$ and from $\hat{\mathbf{X}}_j^{i-1}$ to $\hat{\mathbf{X}}_j^i$ empirically, creating shortcuts that bypass GN blocks in the `Mee` layer. It's worth noting that the intra and inter GN blocks can share parameters across all heights $j$ to accommodate varying heights, or across all `Mee` layers to handle varying layer numbers.

### 3.3 Module Choice and Innovation

`MeGraph` incorporates two fundamental modules: the graph pooling operator and the GN block. This architecture can accommodate any graph pooling method from the `POOL` function family (refer to Sec. 2.2). Furthermore, the GN block is not strictly confined to the graph convolution layer found in standard GCN, GIN, or GAT.

**Graph Pooling.** There are a number of commonly used graph pooling methods, including Diff-Pool [62], TopKPool [22], EdgePool [13], etc. We opt for EdgePool due to its simplicity, efficiency,

and ability to naturally preserve the graph's connectivity through edge contraction. However, edge contraction is applied only to the edges in a specific maximal matching of the graph's nodes [13], thereby setting a lower limit of $50\%$ to the pooling ratio $\eta_v$. This constraint implies that a minimum of $\log_2 N$ pooling operations is required to reduce a graph of $N$ nodes to a single node. To address this limitation, we propose the Stridden EdgePool (S-EdgePool), which allows for a variable pooling stride.

The principle behind S-EdgePool involves dynamically tracking the clusters of nodes created by the contraction of selected edges. Similar to EdgePool, edges are processed in descending order based on their scores. When an edge is contracted, if both ends do not belong to the same node cluster, the two clusters containing the endpoints of the edge merge. The current edge can be contracted if the resulting cluster contains no more than $\tau_c$ nodes after this edge's contraction. The iteration stops prematurely once a pooling ratio, $\eta_v$, is achieved. During pooling, each node cluster is pooled as a new node. When $\tau_c = 2$, S-EdgePool reverts to the original EdgePool. The algorithm's details and pseudocode are available in App. C.2.

For efficiency, we employ the disjoint-set data structure to dynamically maintain the node clusters, which has a complexity of $O(E\alpha(E))$, where $E$ is the number of edges and $\alpha(E)$ is a function that grows slower than $\log(E)$ [50]. The total time complexity of S-EdgePool is equivalent to EdgePool and is calculated as $O(ED+E\log E)$, where $D$ is the embedding size, $O(ED)$ from computing edge scores and $O(E\log E)$ from sorting the edges.

**GN block.** The full GN block, introduced in Sec. 2.1, is implemented as a graph full network (GFuN) layer. This layer exhibits a highly configurable within-block structure, enabling it to express a variety of other architectures (see Sec. 4.2 of [6]), like GCN, GIN, GAT, GatedGCN. Thus, modifying the within-block structure of GFuN is akin to plugging in different GNN cores. Further details can be found in App. C.3.

**Encoder and decoder.** Most preprocessing methods (including positional encodings and graph rewiring), encoding (input embedding) and decoding (readout functions) schemes applicable to GNNs can also be applied to `MeGraph`. We give implementation details in App. C.4.

### 3.4 Computational Complexity and Discussion

The overall complexity of the `MeGraph` model is contingent on the height $h$, the number of `Mee` layers $n$, the chosen modules, and the corresponding hyperparameters. Let $D$ be the embedding size, $V$ the number of nodes, and $E$ the number of edges in the input graph $\mathcal{G}$. The time complexity of S-Edgepool is $O(ED+E\log E)$, and that of a GFuN layer is $O(VD^2+ED)$. Assuming both the pooling ratios of nodes and edges are $\eta$, the total time complexity to construct the mega graph $\mathcal{MG}$ becomes $O((ED + E\log E)/(1-\eta))$, where $\sum_{i=0}^{h-1}\eta^i < 1/(1-\eta)$. Similarly, the total time complexity of an `Mee` layer is $O((VD^2 + ED)/(1-\eta))$. This complexity is equivalent to a typical GNN layer if we consider $1/(1-\eta)$ as a constant (for instance, it is a constant of 2 when $\eta = 0.5$).

Theoretically, when using the same number of layers, `MeGraph` is better at capturing LRIs than standard message-passing GNNs owning to the hierarchical structure (see App. D.1 for details). On the other hand, `MeGraph` can degenerate into standard message-passing GNNs (see App. D.2 for details), indicating it should not perform worse than them on other tasks.

## 4 Experiments

We conduct extensive experiments to evaluate the `MeGraph`'s ability to capture long-range interactions (LRIs) and its performance in general graph learning tasks.

### 4.1 Experimental Settings

**Baselines.** We compare `MeGraph` model to three baselines as follows: 1) `MeGraph` $h$=1 variant does not use the hierarchical structure and falls back to standard GNNs. 2) `MeGraph` $n$=1 variant gives up repeating information exchange over the mega graph. 3) Graph U-Nets [22] uses a U-shaped design and only traverses the multi-scale graphs once.

Due to page limits, statistics of the datasets are provided in App. B.1, hyper-parameters are reported in Table 9, and the training and implementation details are reported in App. E.

Table 1: Results on Graph Theory Benchmark (medium size). For each task, we report the MSE regression loss on test set, averaged over different graph generation methods. Darker blue cells denote better performance and the bold denotes the best one. We provide detailed results on each type of graphs in App. F.7.

| Category | Model | $SP_{sssd}$ | MCC | Diameter | $SP_{ss}$ | ECC | Average |
|---|---|---|---|---|---|---|---|
| Baselines ($h=1$) | $n=1$ | 11.184 | 1.504 | 11.781 | 22.786 | 20.133 | 13.478 |
| | $n=5$ | 3.898 | 1.229 | 5.750 | 12.354 | 18.971 | 8.440 |
| | $n=10$ | 2.326 | 1.264 | 5.529 | 7.038 | 18.876 | 7.006 |
| MeGraph ($h=5$) | $n=1$ | 1.758 | 0.907 | 4.591 | 5.554 | 14.030 | 5.368 |
| EdgePool ($\tau_c=2$) | $n=5$ | 0.790 | 0.767 | 2.212 | 0.712 | 6.593 | 2.215 |
| MeGraph S-EdgePool Variants ($h=5$, $n=5$) | $\tau_c=3$ | 0.660 | 0.747 | 0.719 | 0.459 | **0.942** | 0.705 |
| | $\eta_v=0.3$ | 2.225 | 0.778 | 1.061 | 3.591 | 2.009 | 1.933 |
| | $\eta_v=0.3, \tau_c=4$ | **0.615** | **0.702** | **0.651** | **0.434** | 0.975 | **0.675** |
| | $\eta_v=0.5, \tau_c=4$ | 1.075 | 0.769 | 0.945 | 1.204 | 1.992 | 1.197 |
| | $\eta_v=0.3, \tau_c=4$ (X-Pool) | 0.935 | 0.751 | 0.864 | 1.462 | 2.003 | 1.203 |
| | $\eta_v=0.3, \tau_c=4$ (w/o pw) | 0.632 | 0.730 | 0.864 | 0.765 | 2.334 | 1.065 |
| Graph-UNets | $h=5, n=9, \eta_v=0.3, \tau_c=4$ | 1.118 | 1.008 | 2.031 | 1.166 | 2.584 | 1.581 |

Table 2: Results on Graph Theory Benchmark (large size).

| Category | Model | $SP_{sssd}$ | MCC | Diameter | $SP_{ss}$ | ECC | Average |
|---|---|---|---|---|---|---|---|
| Baseline | h=1,n=5 | 328.014 | 39.4772 | 189.577 | 324.033 | 219.746 | 220.169 |
| MeGraph | h=5,n=5,$\eta_v$=0.3,$\tau_c$=4 | **23.8963** | **16.8321** | **19.2185** | **14.9676** | **44.9234** | **23.9676** |
| Graph-UNets | h=5,n=9,$\eta_v$=0.3,$\tau_c$=4 | 101.009 | 30.3711 | 39.8708 | 100.070 | 75.1185 | 69.2879 |

Table 3: Results on LRGB [17], numbers are taken from corresponding papers. All methods use around 500K parameters for a fair comparison. Message-passing-based models have 5 layers, while the Transformer-based models have 4 layers [17]. PE indicates positional encoding, which makes it easier to distinguish different nodes.

| Methods | Use PE | Peptide-func ↑ | Peptide-struct ↓ |
|---|---|---|---|
| GCN [17] | | 59.30 $_{\pm 0.23}$ | 0.3496 $_{\pm 0.0013}$ |
| GINE [17] | | 55.43 $_{\pm 0.78}$ | 0.3547 $_{\pm 0.0045}$ |
| GatedGCN [17] | | 58.64 $_{\pm 0.77}$ | 0.3420 $_{\pm 0.0013}$ |
| GatedGCN+RWSE [17] | ✓ | 60.69 $_{\pm 0.35}$ | 0.3357 $_{\pm 0.0006}$ |
| GatedGCN+RWSE+VN [10] | ✓ | 66.85 $_{\pm 0.62}$ | 0.2529 $_{\pm 0.0009}$ |
| Transformer+LapPE [17] | ✓ | 63.26 $_{\pm 1.26}$ | 0.2529 $_{\pm 0.0016}$ |
| SAN+LapPE [17] | ✓ | 63.84 $_{\pm 1.21}$ | 0.2683 $_{\pm 0.0043}$ |
| SAN+RWSE [17] | ✓ | 64.39 $_{\pm 0.75}$ | 0.2545 $_{\pm 0.0012}$ |
| GPS [46] | ✓ | 65.35 $_{\pm 0.41}$ | 0.2500 $_{\pm 0.0005}$ |
| MGT+WavePE [43] | ✓ | 68.17 $_{\pm 0.64}$ | **0.2453** $_{\pm 0.0025}$ |
| GNN-AK+ [27] | | 64.80 $_{\pm 0.89}$ | 0.2736 $_{\pm 0.0007}$ |
| SUN [27] | | 67.30 $_{\pm 0.78}$ | 0.2498 $_{\pm 0.0008}$ |
| GraphTrans+PE [27] | ✓ | 63.13 $_{\pm 0.39}$ | 0.2777 $_{\pm 0.0025}$ |
| GINE+PE [27] | ✓ | 64.05 $_{\pm 0.77}$ | 0.2780 $_{\pm 0.0021}$ |
| GINE-MLP-Mixer+PE [27] | ✓ | 69.21 $_{\pm 0.54}$ | 0.2485 $_{\pm 0.0004}$ |
| MeGraph (h=9,n=1) | | 67.52 $_{\pm 0.78}$ | 0.2557 $_{\pm 0.0011}$ |
| MeGraph (h=9,n=4) | | **69.45** $_{\pm 0.77}$ | 0.2507 $_{\pm 0.0009}$ |

## 4.2 Perfomance on LRI Tasks

To test MeGraph's ability to capture long-range interactions, we establish a Graph Theory Benchmark, of which four tasks related to shortest path distance, *i.e.*, Single Source Shortest Path ($SP_{ss}$), Single Source Single Destination Shortest Path ($SP_{sssd}$), Graph Diameter (Diameter) and Eccentricity of nodes (ECC); and 1 task related to connected component, *i.e.*, Maximum Connected Component of the same color (MCC). To generate diversified undirected and unweighted graphs for each task, we adopt the ten methods used in PNA [12] and add four new methods: cycle graph, pseudotree, SBM, and geographic threshold graphs. The details of the dataset generation can be found in App. B.2.

As depicted in Table 1, the MeGraph model with $h=5$, $n=5$ significantly outperforms both the $h=1$ and $n=1$ baselines in terms of reducing regression error across all tasks. It is worth noting that even

Table 4: Results on GNN benchmark. Regression tasks are colored with *blue*. ↓ indicates that smaller numbers are better. Classification tasks are colored with *green*. ↑ indicates that larger numbers are better. Darker colors indicate better performance. † denotes the results are reported in [15].

| Model | ZINC ↓ | AQSOL ↓ | MNIST ↑ | CIFAR10 ↑ | PATTERN ↑ | CLUSTER ↑ |
|---|---|---|---|---|---|---|
| GCN[†] | 0.416 ±0.006 | 1.372 ±0.020 | 90.120 ±0.145 | 54.142 ±0.394 | 85.498 ±0.045 | 47.828 ±1.510 |
| GIN[†] | 0.387 ±0.015 | 1.894 ±0.024 | 96.485 ±0.252 | 55.255 ±1.527 | 85.590 ±0.011 | 58.384 ±0.236 |
| GAT[†] | 0.475 ±0.007 | 1.441 ±0.023 | 95.535 ±0.205 | 64.223 ±0.455 | 75.824 ±1.823 | 57.732 ±0.323 |
| GatedGCN[†] | 0.435 ±0.011 | 1.352 ±0.034 | 97.340 ±0.143 | 67.312 ±0.311 | 84.480 ±0.122 | 60.404 ±0.419 |
| Graph-UNets | 0.332 ±0.010 | 1.063 ±0.018 | 97.130 ±0.227 | 68.567 ±0.339 | 86.257 ±0.078 | 50.371 ±0.243 |
| MeGraph ($h$=1) | 0.323 ±0.002 | 1.075 ±0.007 | 97.570 ±0.168 | 69.890 ±0.209 | 84.845 ±0.021 | 58.178 ±0.079 |
| MeGraph ($n$=1) | 0.310 ±0.005 | 1.038 ±0.018 | 96.867 ±0.167 | 68.522 ±0.239 | 85.507 ±0.402 | 50.396 ±0.082 |
| MeGraph | 0.260 ±0.005 | **1.002** ±0.021 | 97.860 ±0.098 | 69.925 ±0.631 | 86.507 ±0.067 | 68.603 ±0.101 |
| MeGraph$_{best}$ | **0.202** ±0.007 | **1.002** ±0.021 | **97.860** ±0.098 | **69.925** ±0.631 | **86.732** ±0.023 | **68.610** ±0.164 |

Table 5: Results on OGB-G. † indicates that the results are reported in [28].

| Model | molhiv ↑ | molbace ↑ | molbbbp ↑ | molclintox ↑ | molsider ↑ |
|---|---|---|---|---|---|
| GCN[†] | 76.06 ±0.97 | 79.15 ±1.44 | 68.87 ±1.51 | 91.30 ±1.73 | 59.60 ±1.77 |
| GIN[†] | 75.58 ±1.40 | 72.97 ±4.00 | 68.17 ±1.48 | 88.14 ±2.51 | 57.60 ±1.40 |
| Graph-UNets | **79.48** ±1.06 | 81.09 ±1.66 | **71.10** ±0.52 | 91.67 ±1.69 | 59.38 ±0.63 |
| MeGraph ($h$=1) | 78.54 ±1.14 | 71.77 ±2.15 | 67.56 ±1.11 | 89.77 ±3.48 | 58.28 ±0.51 |
| MeGraph ($n$=1) | 78.56 ±1.02 | 79.72 ±1.24 | 67.34 ±0.98 | 91.07 ±2.21 | 58.08 ±0.59 |
| MeGraph | 77.20 ±0.88 | 78.52 ±2.51 | 69.57 ±2.33 | 92.04 ±2.19 | 59.01 ±1.45 |
| MeGraph$_{best}$ | 79.20 ±1.80 | **83.52** ±0.47 | 69.57 ±2.33 | **92.06** ±1.32 | **63.43** ±1.10 |

| Model | moltox21 ↑ | moltoxcast ↑ | molesol ↓ | molfreesolv ↓ | mollipo ↓ |
|---|---|---|---|---|---|
| GCN[†] | 75.29 ±0.69 | 63.54 ±0.42 | 1.114 ±0.03 | 2.640 ±0.23 | 0.797 ±0.02 |
| GIN[†] | 74.91 ±0.51 | 63.41 ±0.74 | 1.173 ±0.05 | 2.755 ±0.34 | 0.757 ±0.01 |
| Graph-UNets | 77.85 ±0.81 | 66.49 ±0.45 | 1.002 ±0.04 | 1.885 ±0.07 | 0.716 ±0.01 |
| MeGraph ($h$=1) | 75.89 ±0.45 | 64.49 ±0.46 | 1.079 ±0.02 | 2.017 ±0.08 | 0.768 ±0.00 |
| MeGraph ($n$=1) | 77.01 ±0.93 | 66.89 ±1.21 | 0.896 ±0.04 | 1.892 ±0.06 | 0.730 ±0.01 |
| MeGraph | **78.11** ±0.47 | 67.67 ±0.53 | 0.886 ±0.02 | **1.876** ±0.05 | 0.726 ±0.00 |
| MeGraph$_{best}$ | **78.11** ±0.47 | **67.90** ±0.19 | **0.867** ±0.02 | **1.876** ±0.05 | **0.688** ±0.01 |

the $h$=5, $n$=1 baseline outperforms the $h$=1, $n$=10 baseline, indicating that adopting a multi-scale graph hierarchy is crucial in these tasks. The improvement is also substantial when compared with our reproduced Graph-UNets using S-EdgePool ([MeGraph] 0.675 vs. [Graph UNets] 1.581). The improvements are more significant when the size of graphs becomes larger (as shown in Table 2). These results collectively demonstrate the superior ability of MeGraph to capture LRIs.

Furthermore, we evaluated MeGraph model and compared it with other recent methods on the Long Range Graph Benchmark (LRGB) [17] that contains real-world tasks that require capturing LRIs. As depicted in Table 3, the $h$=9, $n$=4 variant of MeGraph achieves superior results on the *Peptide-func* task, and comparable performance on the *Peptide-struct* task, relative to state-of-the-art models. It is worth noting that the $n = 1$ variant already surpasses other methods except the recent MLP-Mixer [27] in the *Peptide-func* task.

### 4.3 Generality of MeGraph

To verify the generality of MeGraph model, we evaluate MeGraph on widely adopted GNN Benchmark [15], Open Graph Benchmark [28] and TU Dataset [42]. Results on TU Datasets are available in App. F.3. In addition to the standard model that shares hyper-parameters in similar tasks, we also report MeGraph$_{best}$ with specifically tuned hyper-parameters for each task.

**GNN Benchmark**. We experiment on chemical data (ZINC and AQSOL), image data (MNIST and CIFAR10) and social network data (PATTERN and CLUSTER). As shown in Table 4, MeGraph outperforms the three baselines by a large margin, indicating the effectiveness of repeating both the local and hierarchical information aggregation.

**Open Graph Benchmark (OGB)**. We choose 10 datasets related to molecular graphs from the graph prediction tasks of OGB. The task of all datasets is to predict some properties of molecule graphs

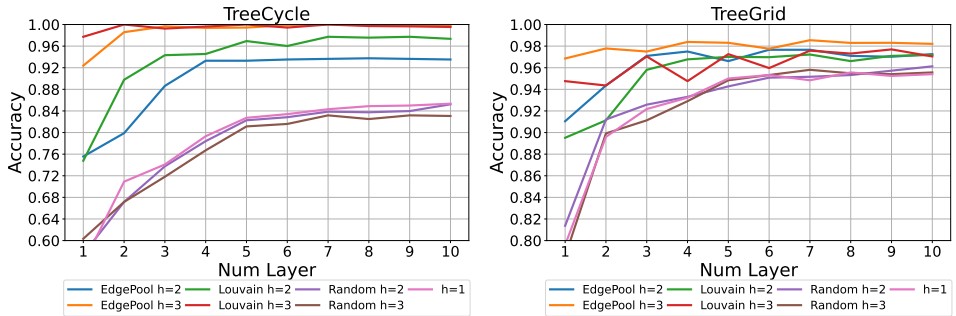

Figure 4: Node classification accuracy (averaged over 10 random repetitions) for `MeGraph` on TreeCycle (left) and TreeGrid (right) datasets by varying the number of `Mee` layers $n$, the height $h$, and the graph pooling methods (EdgePool, Louvain, Random). Clear gaps can be observed among heights 1, 2, and 3 for EdgePool and Louvain [8] methods, while the accuracy is almost invariant among different heights for randomized pooling.

based on their chemical structures. As shown in Table 5, `MeGraph` outperforms the $h$=1 baseline by a large margin, suggesting that building a graph hierarchy is also essential in molecule graphs. The performance of `MeGraph`, $n = 1$ baseline, and the reproduced Graph U-Nets are comparable. This observation may be because the information obtained from multi-hop neighbors offers only marginal improvements compared to the information aggregated hierarchically.

## 4.4 Ablation Study

**Hierarchy vs. Locality.** We study the impact of the height $h$ and the number of `Mee` layers $n$ on four synthetic datasets introduced in [61], which are BAShape, BACommunity, TreeCycle, and TreeGrid. Each dataset contains one graph formed by attaching multiple motifs to a base graph. The motif can be a 'house'-shaped network (BAShape, BACommunity), six-node cycle (TreeCycle), or 3-by-3 grid (TreeGrid). The task is to identify the nodes of the motifs in the fused graph.

As shown in Fig. 4, we can observe clear improvements in performance as the height $h$ and the number of layers $n$ increase when using EdgePool. Although increasing height shows better improvements, both hierarchy and locality are indispensable in TreeCycle and TreeGrid tasks. In App. F.5, we show that the conclusion also holds on BAShape and BACommunity datasets, except that the accuracy is already saturated with height $h = 2$. The significance of integrating locality with hierarchy is also demonstrated in the CLUSTER task, as presented in Table 4. Here, `MeGraph` reaches 68.6% accuracy, which is markedly higher than the 50.4% accuracy achieved by both the $n$=1 baseline and Graph-UNets.

**Varying the Pooling Method.** We varied the graph pooling with two non-learnable pooling methods, Louvain [8] and Random. On the TreeCycle and TreeGrid datasets, as depicted in Fig. 4, Louvain achieves comparable accuracy to EdgePool, while `MeGraph` using random pooling matches the performance of the $h = 1$ variant of `MeGraph`. These observations indicate that: 1) a well-structured hierarchy crafted by suitable graph pooling methods enhances `MeGraph`'s ability to harness hierarchical information; and 2) despite the disruption from a randomly constructed hierarchy, the `MeGraph` model effectively taps into the local structure of the original graph (also discussed in App. D.2).

We further studied the impact of the node pooling ratio $\eta_v$ and the maximum cluster size $\tau_c$ in S-EdgePool by perturbing these parameters. As indicated in Table 1, the best variant ($\eta_v$=0.3,$\tau_c$=4) achieved a regression error about 3x smaller (0.675) compared to the original EdgePool ($\tau_c$=2 with an error of 2.215). This suggests the benefit of having a flexible pooling stride. Moreover, the mega graph produced by S-EdgePool can vary significantly with different parameters. In App. F.2, we visualized the resulting graph hierarchy to illustrate the difference between different S-EdgePool variants. However, irrespective of the parameter set used, `MeGraph` consistently outperforms $h = 1$ baselines. This suggests that `MeGraph` exhibits robustness against different architectures of the mega graph.

**Varying GN Block.** We varied the aggregation function of the GN block as attention (w/ ATT) and gated function (w/ GATE). We observe similar results as in Sec. 4.2 and 4.3, verifying the robustness of `MeGraph` towards different types of GN blocks. Detailed results can be found in Tables 16, 17 and 18 in App. F.6.

**Changing cross update function (X-UPD).** The unpool operation is frequently used by other hierarchical architectures that build upon graph pooling. As illustrated in Table 1, we substituted the *X-Conv* implementation with the *X-Pool* implementation of X-UPD, which resulted in a performance decline from 0.675 to 1.203 (smaller is better). This finding suggests that other hierarchical GNNs might also benefit from replacing the unpool operation with a convolution over the inter-graph edges.

**Disabling bidirectional pathway.** We verify the effectiveness of the bidirectional pathway design by replacing steps 2 and 3 of the Mee layer as a standard message-passing along the inter-graph edges (denoted w/o pw). As shown in Table 1, the performance degrades from 0.675 to 1.065 (smaller is better), which indicates the contribution of the bidirectional pathway.

## 5 Related Work

**Long-Range Interactions (LRIs).** Various methods have been proposed to address the issue of LRIs, including making the GNNs deeper [40]. Another way is to utilize attention and gating mechanism, including GAT [52], jumping knowledge (JK) network [59], incorporating Transformer structures [35, 60, 56, 46] and MLP-Mixer [27]. Another line of research focuses on multi-scale graph hierarchy using graph pooling methods [62, 22, 47], or learning representation based on subgraphs [5, 66]. Recently, Long Range Graph Benchmark [17] has been proposed to better evaluate models' ability to capture LRIs.

**Feature Pyramids and Multi-Scale Feature Fusion.** Multi-scale feature fusion methods on image feature pyramids have been widely studied in computer vision literature, including the U-Net [48], FPN [39], UNet++ [67], and some recent approaches [63, 41, 38, 37]. HRNet [53] is a similar method compared to MeGraph. HRNet alternates between multi-resolution convolutions and multi-resolution fusion by stridden convolutions. However, the above methods are developed for image data. The key difference compared to these approaches is that the multi-scale feature fusion in MeGraph is along the inter-graph edges, which is not as well structured as the pooling operation in image data. For graph networks, the GraphFPN [65] builds a graph feature pyramid according to the image feature pyramid and superpixel hierarchy. It applies GNN layers on the hierarchical graph to exchange information within the graph pyramid. Existing works [22, 20, 47, 30] have also explored similar ideas in graph-structured data. Our approach aligns with the broader concept of multi-scale information fusion, but it is the first method that builds a mega graph using graph pooling operations and alternates local and hierarchical information aggregation.

**Graph Pooling Methods.** Graph pooling is an important part of hierarchical graph representation learning. There have been some traditional graph pooling methods like METIS [32] in early literature. Recently, many learning-based graph pooling methods have been proposed, including the DiffPool [62], TopKPool [22], SAG pool [36], EdgePool [13], MinCutPool [7], Structpool [64], and MEWISPool [44], etc. In this work, we utilize S-EdgePool improved from EdgePool to build the mega graph, while this module can be substituted with any of the above-mentioned pooling methods.

**Graph Neural Network (GNN) Layers.** The GNN layer is the core module of graph representation learning models. Typical GNNs include the GCN [34], GraphSage [24], GAT [52, 9], GIN [58], PNA [12]. MeGraph adopts the full GN block [6] by removing part of links in the module as an elementary block, and similarly this can be replaced by any one of the popular GNN blocks.

## 6 Limitations and Future Work

The MeGraph model suffers from some limitations. The introduced mega graph architecture inevitably increases both the number of trainable parameters and tuneable hyper-parameters. The flexible choices of many modules in MeGraph post burdens on tuning the architecture on specific datasets. For future research, MeGraph encourages new graph pooling methods to yield edge features in addition to node features, when mapping the input graph to the pooled graph. It is also possible to improve MeGraph using adaptive computational steps [49]. Another direction is to apply some expressive but computationally expensive models like Transformers [51] and Neural Logic Machines [14, 57] (only) over the pooled small-sized graphs.

## Acknowledgements

We thank Qizheng He for the discussions on the Graph Theory Dataset, Kirsty You for improving the illustrations, and Jiafei Lv and the Maddison group for their helpful discussions or feedback on the paper draft; This work was supported by the STI 2030-Major Projects under Grant 2021ZD0201404.

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

# Appendix

## Table of Contents

# A   Code and Reproducibility

The code along with the configuration of hyper-parameters to reproduce our experiments can be found at `https://github.com/dhh1995/MeGraph`.

We set the random seed as 2022 for all experiments to enable reproducible results. We provide dataset statistics in Table 6 and details for the proposed graph theory benchmark in Appendix B.2. Details of the hyper-parameters are reported in Table 9. Configuration of all hyper-parameters and the command lines to reproduce the experiments have been included in the code repository.

# B   Dataset Details

## B.1   Dataset Statistics and Metrics

We provide the statistics of all datasets used in our experiments in Table 6 and introduce the evaluation metrics for each dataset.

For Synthetic datasets, we use classification accuracy (ACC) as the evaluation metric. We use Mean Square Error (MSE) as the evaluation metric for all datasets in our Graph Theory Benchmark. For GNN Benchmark, we follow the original work [15] for evaluation, i.e., Mean Absolute Error (MAE) for ZINC and AQSOL, classification accuracy for MNIST and CIFAR10, and balanced classification accuracy for PATTERN and CLUSTER. For OGB Benchmark, we follow the original work [28] and use the ROC-AUC for classification tasks and Root Mean Square Error (RMSE) for regression tasks. For TU datasets, we follow the setting used by [11] and use classification accuracy as the evaluation metric.

## B.2   Graph Theory Benchmark

In this section, we provide the details about the tasks and how the graph features and the labels are generated given a base graph $\mathcal{G} = (\mathcal{V}, \mathcal{E})$:

- Single source single destination shortest path (SP$_{\text{sssd}}$): a source node $s \in \mathcal{V}$ and a destination node $t \in \mathcal{V}$ are selected uniform randomly. The feature of each node $v$ contains three numbers: (1, whether the node $v$ is $s$, whether the node $v$ is $t$). The label of a graph is the length of the shortest path from $s$ to $t$.

- A maximum connected component of the same color (MCC): each node of the graph is colored with one of three colors. The feature for each node is the one-hot representation of its color. The label of graph is the size of the largest connected component of the same color for each color.

- Graph diameter (Diameter): the label of the graph is the diameter of the graph. The diameter of a graph $\mathcal{G}$ is the maximum of the set of shortest path distances between all pairs of nodes in the graph. The feature of each node is a uniform number 1.

- Single source shortest path (SP$_{\text{ss}}$): a source node $s$ is selected uniformly randomly. The feature of each node contains two numbers: (1, whether the node is $s$). The label of each node is the length of the shortest path from $s$ to this node.

- Graph eccentricity (ECC): the label of each node $v$ is node's eccentricity in the graph, which is the maximum distance from $v$ to the other nodes. The feature of each node is a uniform number 1.

For each task and graph generation method, We generate the dataset by the following steps:

- Sample $N$ (number of nodes) from an interval with a total number of graphs. These numbers can be configured. For the medium size setting, the interval is $[20, 50]$, a total of 300 graphs. For the large size setting, the interval is $[100, 200]$, a total of 500 graphs.

- Use the graph generation method to generate a graph of $N$ nodes.

- Create graph features and labels according to the task.

Table 6: The statistics of the datasets used in experiments. Some statistics (like the average number of edges) of the Graph Theory datasets may vary depending on different random graph generation methods. The regression tasks are marked with ✓ in a separate column. The tasks of 4 synthetic datasets are transductive, where the same graph is used for both training and testing. We do not use the node labels as features during the training time. The train-val-test split is over nodes. All other datasets in the table are inductive, where the testing graphs do not occur during training, and the train-val-test split is over graphs.

| Collection | Dataset | # Graphs | Avg # Nodes | Avg # Edges | # Node Feat | # Edge Feat | # Classes | Task | Reg. |
|---|---|---|---|---|---|---|---|---|---|
| Synthetic | BaShape | 1 | 700 | 1761 | 1 | - | 4 | Trans-Node | |
| Synthetic | BaCommunity | 1 | 1400 | 3872 | 10 | - | 8 | Trans-Node | |
| Synthetic | TreeCycle | 1 | 871 | 970 | 1 | - | 2 | Trans-Node | |
| Synthetic | TreeGrid | 1 | 1231 | 1705 | 1 | - | 2 | Trans-Node | |
| GraphTheory | $SP_{sssd}$ | 300 | 35.0 | - | 3 | - | - | Graph | ✓ |
| GraphTheory | Diameter | 300 | 35.0 | - | 1 | - | - | Graph | ✓ |
| GraphTheory | MCC | 300 | 35.0 | - | 3 | - | - | Graph | ✓ |
| GraphTheory | $SP_{ss}$ | 300 | 35.0 | - | 2 | - | - | Node | ✓ |
| GraphTheory | ECC | 300 | 35.0 | - | 1 | - | - | Node | ✓ |
| LRGB | Peptides-func | 15535 | 150.94 | 307.30 | 9 | 3 | 10 | Graph | |
| LRGB | Peptides-struct | 15535 | 150.94 | 307.30 | 9 | 3 | - | Graph | ✓ |
| GNNBenchmark | ZINC | 12000 | 23.16 | 49.83 | 28 | 4 | 2 | Graph | ✓ |
| GNNBenchmark | AQSOL | 9823 | 17.57 | 35.76 | 65 | 5 | 2 | Graph | ✓ |
| GNNBenchmark | MNIST | 70000 | 70.57 | 564.53 | 3 | 1 | 10 | Graph | |
| GNNBenchmark | CIFAR10 | 60000 | 117.63 | 941.07 | 5 | 1 | 10 | Graph | |
| GNNBenchmark | PATTERN | 14000 | 118.89 | 6078.57 | 3 | - | 2 | Node | |
| GNNBenchmark | CLUSTER | 12000 | 117.20 | 4301.72 | 7 | - | 6 | Node | |
| OGB Graph | molhiv | 41127 | 25.51 | 80.45 | 9 | 3 | 2 | Graph | |
| OGB Graph | molbace | 1513 | 34.09 | 107.81 | 9 | 3 | 2 | Graph | |
| OGB Graph | molbbbp | 2039 | 24.06 | 75.97 | 9 | 3 | 2 | Graph | |
| OGB Graph | molclintox | 1477 | 26.16 | 81.93 | 9 | 3 | 2 | Graph | |
| OGB Graph | molsider | 1427 | 33.64 | 104.36 | 9 | 3 | 2 | Graph | |
| OGB Graph | moltox21 | 7831 | 18.57 | 57.16 | 9 | 3 | 2 | Graph | |
| OGB Graph | moltoxcast | 8576 | 18.78 | 57.30 | 9 | 3 | 2 | Graph | |
| OGB Graph | molesol | 1128 | 13.29 | 40.64 | 9 | 3 | - | Graph | ✓ |
| OGB Graph | molfreesolv | 642 | 8.7 | 25.50 | 9 | 3 | - | Graph | ✓ |
| OGB Graph | mollipo | 4200 | 27.04 | 86.04 | 9 | 3 | - | Graph | ✓ |
| TU | MUTAG | 188 | 17.93 | 19.79 | 7 | - | 3 | Graph | |
| TU | NCI1 | 4110 | 29.87 | 32.30 | 37 | - | 2 | Graph | |
| TU | PROTEINS | 1113 | 39.06 | 72.82 | 4 | - | 2 | Graph | |
| TU | D&D | 1178 | 284.32 | 715.66 | 89 | - | 2 | Graph | |
| TU | ENZYMES | 600 | 32.63 | 62.14 | 21 | - | 6 | Graph | |
| TU | IMDB-B | 1000 | 19.77 | 96.53 | 10 | - | 2 | Graph | |
| TU | IMDB-M | 1500 | 13.00 | 65.94 | 10 | - | 3 | Graph | |
| TU | RE-B | 2000 | 429.63 | 497.75 | 10 | - | 2 | Graph | |
| TU | RE-M5K | 4999 | 508.52 | 594.87 | 10 | - | 5 | Graph | |
| TU | RE-M12K | 11929 | 391.41 | 456.89 | 10 | - | 11 | Graph | |

We then provide the details about the random graph generation methods we used to create our Graph Theory datasets.

Following [12], we continue to use undirected and unweighted graphs from a wide variety of types. We inherit their 10 random graph generation methods and quote their descriptions here for completeness (the percentage after the name is the approximate proportion of such graphs in the mixture setting).

- **Erdös-Rényi (ER)** (20%) [18]: with a probability of presence for each edge equal to $p$, where $p$ is independently generated for each graph from $\mathcal{U}[0, 1]$
- **Barabási-Albert (BA)** (20%) [2]: the number of edges for a new node is $k$, which is taken randomly from $\{1, 2, \ldots, N - 1\}$ for each graph
- **Grid** (5%): $m \times k$ 2d grid graph with $N = mk$ and $m$ and $k$ as close as possible
- **Caveman** (5%) [55]: with $m$ cliques of size $k$, with $m$ and $k$ as close as possible
- **Tree** (15%): generated with a power-law degree distribution with exponent 3
- **Ladder graphs** (5%)
- **Line graphs** (5%)
- **Star graphs** (5%)
- **Caterpillar graphs** (10%): with a backbone of size $b$ (drawn from $\mathcal{U}[1, N)$), and $N - b$ pendent vertices uniformly connected to the backbone
- **Lobster graphs** (10%): with a backbone of size $b$ (drawn from $\mathcal{U}[1, N)$), $p$ (drawn from $\mathcal{U}[1, N - b]$ ) pendent vertices uniformly connected to the backbone, and additional $N - b - p$ pendent vertices uniformly connected to the previous pendent vertices.

Additional, we add three more graph generation methods:

- **Cycle graphs**
- **Pseudotree graphs**: A tree graph plus an additional edge. The graph is generated by first generating a cycle graph of size $m = \text{sample}(0.3N, 0.6N)$. Then $n - m$ remaining nodes are sampled to $m$ parts, where $i$-th part represents the size of the tree hanging on the $i$-th node on the cycle. The trees are randomly generated with the given size.
- **Stochastic Block Model (SBM) graphs**: graphs with clusters. We randomly sample the size of each block to be random from (5, 15), and the probability of edge within the block to be random from (0.3, 0.5) and those for other edges to be random from (0.005, 0.01). To make all the tasks well-defined, we filtered out the unconnected graphs during the generation.
- **Geographic (Geo) graphs**: geographic threshold graphs, but with added edges via a minimum spanning tree algorithm, to ensure all nodes are connected. This graph generation method is introduced by [6] in their codebase [2]. We use the geographic threshold $\theta = 200$ instead of the default value $\theta = 1000$.

Note that we do not have randomization after the graph generation as in [12]. Therefore, very long diameter is preserved for some type of graphs.

## C   Method Details

### C.1   Cross Update Function

The cross update function $(\mathbf{X}'_j, \hat{\mathbf{X}}'_j, \mathbf{X}'_{j+1}) = \texttt{X-UPD}(j, \mathbf{X}_j, \hat{\mathbf{X}}_j, \mathbf{X}_{j+1})$ perform information exchange in consecutive hierarchies.

The *X-Conv* realization contains the following steps:

1. Merge the node features of $\mathbf{X}_j$ and $\mathbf{X}_{j+1}$ with the inter-graph feature $\hat{\mathbf{X}}_j$, results in $\bar{\mathbf{X}}_j$.
2. Apply GN blocks on inter-graph $\hat{\mathcal{G}}_j$: $\hat{\mathbf{X}}'_j = \text{GN}^{i,j}_{\text{inter}}(\hat{\mathcal{G}}_j, \bar{\mathbf{X}}_j)$.
3. Retrieve $\mathbf{X}'_j$ and $\mathbf{X}'_{j+1}$ from the node features of inter-graph features $\hat{\mathbf{X}}'_j$.

---

[2]https://github.com/deepmind/graph_nets, the shortest path demo

## C.2 S-EdgePool

In this subsection, we introduce the details of S-EdgePool. We first introduce the score generation method, then give details about the SELECT, CONNECT, REDUCE and EXPAND functions, and lastly provide pseudocode of the algorithm.

### C.2.1 Edge Score Generation

Both S-EdgePool and EdgePool methods compute a raw edge score $\mathbf{r}_k$ for each edge $k$ using a linear layer:

$$\mathbf{r}_k = \mathbf{W} \cdot (\mathbf{V}_{s_k} || \mathbf{V}_{t_k} || \mathbf{E}_k) + \mathbf{b}$$

where $s_k$ and $t_k$ are the source and target nodes of edge $k$, $\mathbf{V}$ is node features, $\mathbf{E}$ is edge features, $\mathbf{W}$ and $\mathbf{b}$ are learned parameters. The raw edge scores are further normalized by a local softmax function over all edges of a node:

$$\mathbf{w}_k = \exp(\mathbf{r}_k) / \sum_{k', t_{k'} = t_k} \exp(\mathbf{r}_{k'}),$$

and biased by a constant 0.5 [13].

### C.2.2 Select, Connect, Reduce and Expand

SELECT **step.** S-EdgePool shares the same computations as in EdgePool to generate learnable edge scores, as detailed above. Then, we use a clustering procedure to determine the subset of nodes to be reduced.

Let $I_\mathbf{v}$ be the identifier of the cluster containing a set of nodes $\mathbf{v}$. Initially, we let $\mathbf{v} = \{v\}$ for every single node $v$. A contraction of an edge merges a pair of nodes $(v, v')$ connected by this edge (where $v \in \mathbf{v}$, $v' \in \mathbf{v}'$ and $\mathbf{v} \neq \mathbf{v}'$), and thus unifies the cluster identifiers, i.e., $I_\mathbf{v} = I_{\mathbf{v}'} = I_{\mathbf{v}_{\text{merge}}}$ and $\mathbf{v}_{\text{merge}} = \mathbf{v} \cup \mathbf{v}'$. That is, once an edge connecting any pair of nodes from two distinct clusters is contracted, we merge the two clusters and unify their identifiers. Edges are visited sequentially by a decreasing order on the edge scores, and contractions are implemented if valid. We set the maximum size of the node clusters to be a parameter $\tau_c$, where $\tau_c = 2$ degenerates to the case of EdgePool [13]. We further introduce the pooling ratio $\eta_v$ to control the minimal number of remaining clusters after edge contractions to be $N^v * \eta_v$. Contractions that violate the above two constraints are invalid and will be skipped. Both parameters control the number of nodes in the pooled graph. In our implementation, the cluster of nodes is dynamically maintained using the disjoint-set data structure [21].

Then each node cluster $i$ collapses into a new node $\tilde{v}$ of the pooled graph (i.e. $S_{\tilde{v}} = \{v | I_v = i\}$), with inter-graph edges connect the nodes in the cluster to the new node $\tilde{v}$.

CONNECT **step.** The CONNECT function rebuilds the edge set $\tilde{\mathcal{E}}$ between the nodes in $\tilde{\mathcal{V}}$. As aforementioned, we build the pooled graph's nodes according to node clusters. We call this mapping function from node clusters to new nodes as *c2n*. After that, we build the pooled graph's edges following three steps: First, for all edges in the original graph, we find out the corresponding node cluster(s) of its two endpoints (using a disjoint-set's find index operation). Then, we find out the corresponding new nodes by using the mapping function $n$. Last, we add a new edge between the new nodes.

REDUCE **and** EXPAND **step.** The REDUCE and EXPAND are generalized from the method mentioned in [13]. The REDUCE function computes new node features and edge features. We follow their method to compute new node features by taking the sum of the node features and multiplying it by the edge score. Specifically, we generalize the computation between two nodes to a node cluster. The node clusters are maintained with a disjoint-set data structure and a cluster $S_{\tilde{v}}$ consists of $|S_{\tilde{v}}|$ nodes. We define $\mathcal{E}_{\tilde{v}}^{ds}$ as a set of $|S_{\tilde{v}}| - 1$ edges, where the edges are the selected edges to be contracted in the SELECT step. Then,

$$c_{\tilde{v}} = \frac{1 + \sum_{e_k \in \mathcal{E}_{\tilde{v}}^{ds}} \mathbf{w}_k}{|S_{\tilde{v}}|}$$

$$\mathbf{V}_{\tilde{v}} = c_{\tilde{v}} \sum_{v \in S_{\tilde{v}}} \mathbf{V}_v$$

To integrate the edge features between two node clusters, we first find all the connected edges between the two node clusters (the edges between node clusters are edges that connect two nodes from different node clusters). Then, we use the sum of all the connected edges' features between the two node clusters as the new edge's features.

The `EXPAND` function is also referred as *unpool* operation. It computes node features of the input graph $\mathbf{V}_v$ given the node features of the pooled graph $\mathbf{V}_{\tilde{v}}$ as following:

$$\mathbf{V}_v = \frac{\mathbf{V}_{\tilde{v}}}{c_{\tilde{v}}}$$

### C.2.3 Pseudo Code

The pseudo-code includes two parts, where Algorithm 1 describes how to maintain the clusters using a disjoint-set data structure, and Algorithm 2 describes the procedure of S-EdgePool that generates a pooled graph $\tilde{\mathcal{G}}$ with configurable node pooling ratio $\eta_v$ and maximum of cluster sizes $\tau_c$.

---

**Algorithm 1** Get Cluster Index And Cluster Size of a Node (Using disjoint-set data structure)

---

    **function** InitializeDisjointSet(graph $\mathcal{G}(\mathcal{V}, \mathcal{E})$)
      **for** $v \in \mathcal{V}$ **do**
        $index[v] = v$ {the identifier of the cluster the node $v$ belongs to}
      **end for**
    **end function**
    **function** FindIndex(node $v$)
      **if** $index[v] = v$ **then**
        **return** $v$
      **else**
        $index[v] \leftarrow$ FindIndex($index[v]$)
        **return** $index[v]$
      **end if**
    **end function**
    **function** FindIndexAndSize(node $v$)
      $i \leftarrow$ FindIndex($v$)
      $s \leftarrow size[i]$
      **return** $i, s$
    **end function**
    **function** MERGE(cluster index $x$, cluster index $y$)
      $size[y] \leftarrow size[x] + size[y]$
      $index[x] \leftarrow index[y]$
    **end function**

---

### C.3 GFuN

We first realize the $\phi_e$, $\phi_v$, $\phi_u$ functions in the full GN block (Sec 2.1 and [6]) as neural networks:

$$\begin{align}
\mathbf{E}'_k &= \text{NN}_e(\mathbf{E}_k, \mathbf{V}_{s_k}, \mathbf{V}_{t_k}, \mathbf{u}), \tag{2}\\
\mathbf{V}'_i &= \text{NN}_v(\bar{\mathbf{E}}'_i, \mathbf{V}_i, \mathbf{u}), \tag{3}\\
\mathbf{u}' &= \text{NN}_u(\bar{\mathbf{E}}', \bar{\mathbf{V}}', \mathbf{u}), \tag{4}
\end{align}$$

respectively, where

$$\begin{align}
\bar{\mathbf{E}}'_i &= \rho^{e \to v}(\{\mathbf{E}'_k\}_{k \in [1...N^e], t_k = i}), \tag{5}\\
\bar{\mathbf{E}}' &= \rho^{e \to u}(\mathbf{E}'), \tag{6}\\
\bar{\mathbf{V}}' &= \rho^{v \to u}(\mathbf{V}'). \tag{7}
\end{align}$$

We further decompose the neural networks according to the features in the function:

$$\begin{align}
\text{NN}_e(\mathbf{E}_k, \mathbf{V}_{s_k}, \mathbf{V}_{t_k}, \mathbf{u}) &= \text{NN}_{e \leftarrow e}(\mathbf{E}_k) + \text{NN}_{e \leftarrow v_s}(\mathbf{V}_{s_k}) + \text{NN}_{e \leftarrow v_t}(\mathbf{V}_{t_k}) + \text{NN}_{e \leftarrow u}(\mathbf{u}), \tag{8}\\
\text{NN}_v(\bar{\mathbf{E}}'_i, \mathbf{V}_i, \mathbf{u}) &= \text{NN}_{v \leftarrow e}(\bar{\mathbf{E}}'_i) + \text{NN}_{v \leftarrow v}(\mathbf{V}_i) + \text{NN}_{v \leftarrow u}(\mathbf{u}), \tag{9}\\
\text{NN}_u(\bar{\mathbf{E}}', \bar{\mathbf{V}}', \mathbf{u}) &= \text{NN}_{u \leftarrow e}(\bar{\mathbf{E}}') + \text{NN}_{u \leftarrow v}(\bar{\mathbf{V}}') + \text{NN}_{u \leftarrow u}(\mathbf{u}) \tag{10}
\end{align}$$

---

**Algorithm 2** Strided EdgePool

---

**input** graph $\mathcal{G} = (\mathcal{V}, \mathcal{E})$, edge scores $\mathbf{w}$, node pooling ratio $\eta_v$, maximum cluster sizes $\tau_c$.
**output** pooled graph $\tilde{\mathcal{G}} = (\tilde{\mathcal{V}}, \tilde{\mathcal{E}})$ and inter graph $\hat{\mathcal{G}} = (\hat{\mathcal{V}}, \hat{\mathcal{E}})$
  InitializeDisjointSet($\mathcal{G}$)
  $remains \leftarrow N^v$ {$N^v$ is the number of nodes in graph $\mathcal{G}$}
  $\bar{\mathcal{E}} \leftarrow$ Sort the edges $\mathcal{E}$ according to the edge scores $\mathbf{w}$ decreasingly.
  **for** $e \in \bar{\mathcal{E}}$ **do**
    $x, y \leftarrow$ the two endpoints of the edge $e$
    $rx, sx \leftarrow$ FindIndexAndSize($x$)
    $ry, sy \leftarrow$ FindIndexAndSize($y$)
    **if** $rx \neq ry$ and $(sx + sy \leq \tau_c)$ **then**
      Merge($x, y$)
      $remains \leftarrow remains - 1$
      **if** $remains \leq N^v * \eta_v$ **then**
        **break**
      **end if**
    **end if**
  **end for**
  $\tilde{\mathcal{V}}, \tilde{\mathcal{E}}, \hat{\mathcal{V}}, \hat{\mathcal{E}} \leftarrow \{\}, \{\}, \{\}, \{\}$
  create empty mapping *c2n* from cluster index to nodes
  **for** $v \in \mathcal{V}$ **do**
    **if** FindIndex($v$) $= v$ **then**
      create new node $\tilde{v}$
      $c2n[v] = \tilde{v}$
      $\tilde{\mathcal{V}} \leftarrow \tilde{\mathcal{V}} \cup \{\tilde{v}\}$
    **end if**
  **end for**
  **for** $e \in \mathcal{E}$ **do**
    $x, y \leftarrow$ the two endpoints of the edge $e$
    $\tilde{x} \leftarrow c2n[\text{FindIndex}(x)]$
    $\tilde{y} \leftarrow c2n[\text{FindIndex}(y)]$
    $\tilde{\mathcal{E}} \leftarrow \tilde{\mathcal{E}} \cup \{(\tilde{x}, \tilde{y})\}$
  **end for**
  **for** $v \in \mathcal{V}$ **do**
    $\tilde{v} \leftarrow c2n[\text{FindIndex}(v)]$
    $\hat{\mathcal{E}} \leftarrow \hat{\mathcal{E}} \cup \{(v, \tilde{v})\}$
  **end for**
  $\hat{\mathcal{V}} \leftarrow \mathcal{V} \cup \tilde{\mathcal{V}}$

---

However, such GN block uses 10 times the number of parameters as the standard GCN [34] layer when the node, edge and global embedding dimensions are all equivalent. In practice, we disable all computations related to global features $\mathbf{u}$, as well as the neural networks $\text{NN}_{e \leftarrow e}$ and $\text{NN}_{e \leftarrow v_t}$. We also set $\text{NN}_{v \leftarrow e}$ to be Identity.

In practice, we use the summation function as the aggregator function $\rho^{e \rightarrow v}$ by default. But other choices like MEAN, MAX, gated summation, attention or their combinations can also be used.

Overall, we call such GN block as graph full network (GFuN).

### C.4 Encorder and Decoder

**Encoder**. For input embedding, we use the Linear layer or Embedding layer to embed input features. For example, we follow [15] and use the Linear layer on MNIST and CIFAR10 datasets, and use the Embedding layer on ZINC and AQSOL datasets. For the molecular graph in OGB, we use the same embedding method as in the original work [28]. Besides, we can adopt positional encoding methods like Laplacian [15] and Random Walk [16] to further embed global and local graph structure information. The embedding of positional encoding can be combined into (like concatenation, addition, etc.) input features and form new embeddings.

**Decoder**. We can freely choose from the multi-scale features computed during the *process* stage as inputs to the decoder module. Empirically, we use the features on the original graph for prediction in all experiments. For node-level tasks, we apply a last GNN layer on the original graph to get logits for every node. For graph-level tasks, we first use global pooling functions to aggregate features. We can use common global pooling methods like SUM, MEAN, MAX, or their combination. After the global pool, we use MLP layer(s) to generate the prediction.

### C.5 Architecture Variants

We can replace some GN blocks within `Mee` layers as an Identity block to reduce the time complexity. We call the height $j$ is reserved if the intra GN block of height $j$ is not replaced by an Identity block. We prefer to reserve an interval of consecutive heights for the `Mee` layers. (The inter GN blocks between these heights remain unchanged while others are replaced as identities) By varying the heights reserved in each `Mee` layers, we can create a large number of variants of `MeGraph` model including U-Shaped, Bridge-Shaped and Staircase-Shaped.

**U-Shaped**. This variant is similar to Graph U-Net [22]. In this U-Shaped variant, the relationship between the number of layers $n$ and height $h$ is $n = 2h + 1$, and there is only one GN block in each layer. We keep the GN block at height $j = i$ for each layer $i$ at the first half layers and keep the GN block at height $j = n - i + 1$ for each layer $i$ at the later half layers. In the middle layer, only the last height $j = h = (n-1)/2$ has a GN block.

**Bridge-Shaped**. In this variant, all GN blocks are combined like an arch bridge. Describe in detail, in the first and last layers, there are GN blocks in each height. In other layers, there are GN blocks at a height of 1 to $j$ (where $1 < j < h$).

**Staircase-Shaped**. There are four forms in this variant, and the number of layers $n$ is equal to the height $h$ in all forms. The first form is like the 'downward' staircase. In each layer $i$ of this form, there are GN blocks at the height of $j$ to $h$ (where $j = i$). The second form is the inverted first form. In each layer $i$ of this second form, there are GN blocks at height of 1 to $h - i + 1$ (where $j = i$). The last two forms are the mirror of the first and second forms.

## D    Theoretical Discussions

### D.1    Smaller Number of Aggregation Steps for Capturing Long-Range Interactions

We rephrase the analysis provided in [47] as following:

We analyze the number of aggregation steps required to capture long-range interactions between nodes in the original graph while assuming the node representation capacity is large enough.

Standard message-passing GNNs require $n$ aggregation steps to capture long-range interactions of $n$ hops away, therefore requiring a stack of $n$ layers, which could be expensive when $n$ is large.

We also assume the height $h$ of the hierarchy is large enough so that all nodes of the original graph are pooled into a single node. In that case, the information aggregation along the hierarchy captures all pairs of LRIs into the embedding of the single node. Which means the number of aggregation steps of `MeGraph` is $h$. When we adopt a pooling method that coarsens the graph at least half, $h$ is at most $O(\log(|V|))$ where $|V|$ is the number of nodes of the input graph. Therefore, the height $h$ is significantly smaller than the diameter of the graph (which could be $O(|V|)$) in most cases.

### D.2    `MeGraph` can degenerate to standard GNNs

`MeGraph` can learn a gating function (within the X-UPD function) that only reserves the features of the same scale while performing cross-scale information exchanging. In that case, there will be no information exchange across multi-scale graphs, and features other than those in the original scale will not be aggregated. We provide a proof sketch below, while the results of the random pooling method ablation study in Sec. 4.4 also provide empirical evidence.

**Proof**: The cross update function is $(X'_j, \hat{X}'_j, X'_{j+1}) = \texttt{X-UPD}(j, X_j, \hat{X}_j, X_{j+1})$. There is a residual function applied here, and we assume it is implemented as a gated residual: $X''_j = \sigma(\alpha)X_j + \sigma(\beta)X'_j$, where $\sigma$ is the sigmoid function and $\alpha, \beta$ are learnable parameters. Theoremetically, it is possible that

Table 7: Running time (s) for one epoch on the GNN benchmark. See Sec. E for more implementation details.

| | ZINC | AQSOL | CIFAR10 | MNIST | PATTERN | CLUSTER |
|---|---|---|---|---|---|---|
| Megraph ($h = 5$) | 25.69 | 20.22 | 336.63 | 307.23 | 101.52 | 69.65 |
| Megraph ($h = 1$) | 2.41 | 1.67 | 51.74 | 38.60 | 9.21 | 6.52 |

Table 8: Running time (s) for one epoch on the OGBG datasets. See Sec. E for more implementation details.

| | molhiv | molbace | molbbbp | molclintox | molsider |
|---|---|---|---|---|---|
| Megraph ($h = 5$) | 393.42 | 14.70 | 20.36 | 14.03 | 14.12 |
| Megraph ($h = 1$) | 22.50 | 1.43 | 1.58 | 1.26 | 1.41 |

| | moltox21 | moltoxcast | molesol | molfreesolv | mollipo |
|---|---|---|---|---|---|
| Megraph ($h = 5$) | 70.15 | 78.77 | 11.68 | 6.24 | 44.77 |
| Megraph ($h = 1$) | 5.27 | 8.17 | 0.76 | 0.41 | 2.77 |

$\sigma(\alpha) = 1$ and $\sigma(\beta) = 0$ after training. In that case, $X_j'' = X_j$, which means $X_j$ is not changed over steps 2 and 3 of the Mee layer. Therefore, $X_0^i = GN_{intra}^{i,0}(\mathcal{G}, X_0^{i-1})$, this is equivalent to a simple GNN layer that $X^i = GNN_i(\mathcal{G}, X^{i-1})$ as $X_0^i$ is the features of the original graph and $GN_{intra}$ is a GNN layer. Therefore, `MeGraph` degenerates to standard message-passing GNNs in this case. ∎

## E   Implementation and Training Details

We use PyTorch [45] and Deep Graph Library (DGL) [54] to implement our method.

We implement S-EdgePool using DGL, extending from the original implementation of EdgePool in the Pytorch Geometric library (PYG) [19]. We did Constant optimization over the implementation to speed up the training and inference of the pooling. We further use Taichi-Lang [29] to speed up the dynamic node clustering process of S-EdgePool. The practical running time of `MeGraph` model with height $h > 1$ after optimization is about $2h$ times as the $h = 1$ baseline. This is still slower than the theoretical computational complexity due to the constant in the implementation and the difficulty of paralleling the sequential visitation of edges (according to their scores) in the EdgePool and S-EdgePool. This process could be further speed up by implementing the operations with the CUDA library. We provide the practical running time for $h > 1$ and $h = 1$ in GNN benchmark and OGB-G datasets in Tables 7 and 8. The speed is slower than the theoretical one partially due to the pooling ratio of edges being larger than the nodes, making the number of edges decrease slowly over the hierarchy. To further speed up, we could use the variants of `MeGraph` introduced in App. C.5 by skipping some computation modules.

We run all our experiments on V100 GPUs and M40 GPUs. For training the neural networks, we use Adam [33] as the optimizer. We report the hyper-parameters of the `Megraph` in Table 9.

For models using GFuN layer as the core GN block, we find it benefits from using layer norms [4]. However, for models using GCN layer as the core GN block, we find it performs best when using batch norms [31].

## F   Additional Experiment Results

### F.1   Experimental Protocol

We evaluate `MeGraph` on public real-world graph benchmarks. To fairly compare `MeGraph` with the baselines, we use the following experimental protocols. We first report the public baseline results and our reproduced standard GCN's results. We then replace GCN layers with GFuN layers (which is equivalent to `MeGraph` ($h = 1$)) to serve as another baseline. We tune the hyper-parameters (such as learning rate, dropout rate and the readout global pooling method, etc.) of `MeGraph` ($h = 1$) and choose the best configurations. We then run other diversely configured `MeGraph` candidates by tuning

Table 9: Hyper-parameters of the standard version of `MeGraph` for each dataset. It is worth noting that the total number of GNN layers is equals to one plus the number of `Mee` layers as $n + 1$.

| Hyper-parameters | Synthetic Datasets | Graph Theory Benchmark | LRGB Benchmark | GNN Benchmark | OGB Benchmark | TU Datasets |
|---|---|---|---|---|---|---|
| Repeated Runs | 10 | 5 | 4 | 4 | 5 | 1 for each fold |
| Epochs per run | 200 for BA* 500 for Tree* | 300 (200 for MCC) | 200 | 200 (100 for MNIST, CIFRA10) | 100 | 100 (200 for ENZYMES) |
| Learning rate | 0.002 | 0.002 (0.005 for MCC) | 0.001 | 0.001 | 0.001 | 0.002 |
| Weight decay | 0.0005 | 0.0005 | 0 | 0 | 0.0005 | 0.0005 |
| Node hidden dim | 64 | 128 | 160 | 144 | 300 | 128 |
| Edge hidden dim (for GFuN) | 64 | 128 | 160 | 144 | 300 | 128 |
| Num `Mee` layers $n$ | - | - | 4 | 3 | 4 | 2 |
| Height $h$ | - | - | 9 | 5 | 5 | 3 or 5 |
| Batch size | 32 | 32 | 128 | 128 | 32 | 128 |
| Input embedding | False | True | True | True | True | True |
| Global pooling | Mean | Mean Max | Mean Max Sum | Mean | Mean | Mean Max Sum |
| Dataset split (train:val:test) | 8:1:1 | 8:1:1 | Original split | Original split | Original split | 10-fold cross validation |

other hyper-parameters that only matters for $h > 1$, and these hyper-parameters are referred to as the `MeGraph` hyper-parameters. Detailed configurations are put in Table 9 in App. E.

## F.2 Visualization

We plotted the graph hierarchies discovered by `MeGraph` in the shortest path tasks of Graph Theory Benchmark. In Figure 5, the S-EdgePool with $\eta_v = 0.3, \tau_c = 4$ well preserves the structure of the graph after pooling, while the S-EdgePool with $\eta_v = 0.3$ (no cluster size limit) sometimes pooled too many nodes together, breaking the graph structure. The former S-EdgePool leads to better performance as indicated in Table 1. We also plot the hierarchy for SBM-generated graphs in Figure 6, indicating that EdgePool can handle graphs that naturally contains clusters of different size.

## F.3 Other Real-Wrold Datasets

**TU dataset** consists of over 120 datasets of varying sizes from a wide range of applications. We choose 10 datasets, 5 of which are molecule datasets (MUTAG, NCI1, PROTEINS, D&D and ENZYMES) and the other 5 are social networks (IMDB-B, IMDB-M, REDDIT-BINARY, REDDIT-MULTI-5K and REDDIT-MULTI-12K). They are all graph classification tasks. For more details of each dataset, please refer to the original work [42].

Our `Megraph` uses the same network structure and hyper-parameters for the same type of dataset. As shown in Table 10, our `Megraph` achieves about $1\%$ absolute gain than the $h = 1$ Baselines.

## F.4 GFuN

We show our GFuN results on real-world datasets compared to our reproduced GCN in Table 11, 12 and 13. Both GCN and GFuN have the same hyper-parameters except the batch norm for GCN and layer norm for GFuN as stated in Appendix E.

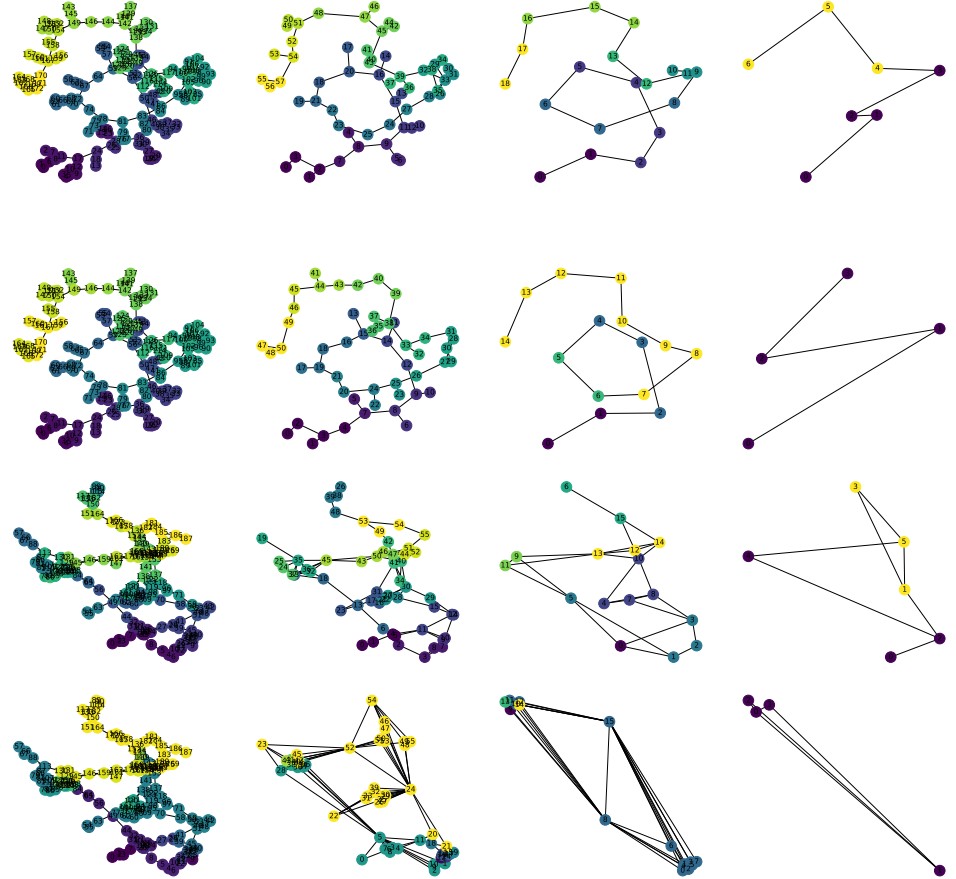

Figure 5: The visualization of the resulting graph hierarchy of `MeGraph`. Each row is a graph hierarchy, and the nodes with the same color in the same graph are in the same cluster and are pooled into a single node in the next hierarchy. The 1st, 3rd hierarchies use S-EdgePool with $\eta_v = 0.3, \tau_c = 4$ and the 2nd and 4th hierarchies use S-EdgePool with $\eta_v = 0.3$. The graph in the 1st and 2nd hierarchies is generated by the 'lobster' method, the graph in the 3rd and 4th hierarchies is generated by the 'geo' method. The task is to compute the length of the shortest path between two nodes.

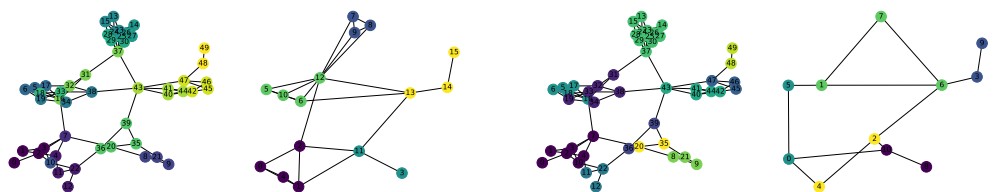

Figure 6: The graph hierarchy visualization, using S-EdgePool with $\eta_v = 0.3, \tau_c = 4$ (left two figures) and Louvain pooling (right two figures) on the same SBM generated graph. The task is the shortest path.

## F.5 Synthetic Datasets

Figures 7 and 8 show the influence of the height $h$ and the number of `Mee` layers $n$ for `MeGraph` model on the BAShape and BACommunity datasets. The trends are similar while less significant on easier datasets BAShape and BACommunity.

## F.6 Varying GN block

We vary the aggregation function of the GN block as attention (w/ ATT) and gated function (w/ GATE). We observe similar results as in Sec. 4.2 and 4.3, verifying the robustness of `MeGraph` over different GN blocks. Results are shown in Tables 16, 17 and 18.

Table 10: Results on Tu Dataset. † means the results taken from [11] (*: The result of GCN on ENZYMES is 100 epoch).

| Model | MUTAG ↑ | NCI1 ↑ | PROTEINS ↑ | D&D ↑ | ENZYMES ↑ | Average |
|---|---|---|---|---|---|---|
| GCN† | 87.20 ±5.11 | 83.65 ±1.69 | 75.65 ±3.24 | 79.12 ±3.07 | 66.50 ±6.91* | 78.42 |
| GIN† | 89.40 ±5.60 | 82.70 ±1.70 | 76.20 ±2.80 | - | - | - |
| GCN | 92.46 ±6.55 | 82.55 ±0.99 | 77.82 ±4.52 | 80.56 ±2.40 | 74.17 ±5.59 | 81.51 |
| MeGraph ($h$=1) | 93.01 ±6.83 | 82.53 ±1.89 | 81.32 ±4.08 | 81.32 ±3.17 | 74.83 ±3.20 | 82.60 |
| MeGraph | 93.07 ±6.71 | 83.99 ±0.98 | 81.41 ±3.10 | 81.24 ±2.39 | **75.17** ±4.86 | 82.98 |
| MeGraph$_{best}$ | **94.12** ±5.02 | **84.40** ±1.11 | **81.68** ±3.40 | **82.00** ±2.86 | **75.17** ±4.86 | **83.47** |

| Model | IMDB-B ↑ | IMDB-M ↑ | RE-B ↑ | RE-M5K ↑ | RE-M12K ↑ | Average |
|---|---|---|---|---|---|---|
| GCN | **76.00** ±3.44 | 50.33 ±1.89 | 91.15 ±1.63 | 56.47 ±1.54 | 48.71 ±0.88 | 64.53 |
| MeGraph ($h$=1) | 68.60 ±3.53 | 51.33 ±2.23 | 93.10 ±1.16 | 57.47 ±2.31 | 51.56 ±1.06 | 64.41 |
| MeGraph | 72.40 ±2.80 | 51.27 ±2.71 | **93.75** ±1.25 | 57.69 ±2.22 | 52.03 ±0.86 | 65.43 |
| MeGraph$_{best}$ | 74.30 ±2.97 | **52.00** ±2.49 | **93.75** ±1.25 | **58.45** ±2.22 | **52.13** ±1.01 | **66.13** |

Table 11: Comparison between GCN and GFuN on GNN benchmark.

| Model | ZINC ↓ | AQSOL ↓ | MNIST ↑ | CIFAR10 ↑ | PATTERN ↑ | CLUSTER ↑ |
|---|---|---|---|---|---|---|
| GCN | 0.426 ±0.015 | 1.397 ±0.029 | 90.140 ±0.140 | 51.050 ±0.390 | 84.672 ±0.054 | 47.541 ±0.940 |
| GFuN | 0.364 ±0.003 | 1.386 ±0.024 | 95.560 ±0.190 | 61.060 ±0.500 | 84.845 ±0.021 | 58.178 ±0.079 |
| MeGraph ($h$=1) | 0.323 ±0.002 | 1.075 ±0.007 | 97.570 ±0.168 | 69.890 ±0.209 | 84.845 ±0.021 | 58.178 ±0.079 |
| MeGraph | 0.260 ±0.005 | 1.002 ±0.021 | 97.860 ±0.098 | 69.925 ±0.631 | 86.507 ±0.067 | 68.603 ±0.101 |

Table 12: Comparison between GCN and GFuN on OGB-G.

| Model | molhiv ↑ | molbace ↑ | molbbbp ↑ | molclintox ↑ | molsider ↑ |
|---|---|---|---|---|---|
| GCN | 75.40 ±1.29 | 76.01 ±3.31 | 67.35 ±0.96 | 89.62 ±2.27 | 58.08 ±0.78 |
| GFuN | 78.54 ±1.14 | 71.77 ±2.15 | 67.56 ±1.11 | 89.77 ±3.48 | 58.28 ±0.51 |
| MeGraph ($h$=1) | 78.54 ±1.14 | 71.77 ±2.15 | 67.56 ±1.11 | 89.77 ±3.48 | 58.28 ±0.51 |
| MeGraph | 77.20 ±0.88 | 78.52 ±2.51 | 69.57 ±2.33 | 92.04 ±2.19 | 59.01 ±1.45 |

| Model | moltox21 ↑ | moltoxcast ↑ | molesol ↓ | molfreesolv ↓ | mollipo ↓ |
|---|---|---|---|---|---|
| GCN | 75.11 ±0.41 | 64.13 ±0.52 | 1.141 ±0.02 | 2.407 ±0.15 | 0.788 ±0.01 |
| GFuN | 75.89 ±0.45 | 64.49 ±0.46 | 1.079 ±0.02 | 2.017 ±0.08 | 0.768 ±0.00 |
| MeGraph ($h$=1) | 75.89 ±0.45 | 64.49 ±0.46 | 1.079 ±0.02 | 2.017 ±0.08 | 0.768 ±0.00 |
| MeGraph | 78.11 ±0.47 | 67.67 ±0.53 | 0.886 ±0.02 | 1.876 ±0.05 | 0.726 ±0.00 |

Table 13: Comparison between GCN and GFuN on Tu Dataset.

| Model | MUTAG ↑ | NCI1 ↑ | PROTEINS ↑ | D&D ↑ | ENZYMES ↑ | Average |
|---|---|---|---|---|---|---|
| GCN | 92.46 ±6.55 | 82.55 ±0.99 | 77.82 ±4.52 | 80.56 ±2.40 | 74.17 ±5.59 | 81.51 |
| GFuN | 93.01 ±7.96 | 82.80 ±1.30 | 80.60 ±3.83 | 82.43 ±2.60 | 73.00 ±5.31 | 82.37 |

| Model | IMDB-B ↑ | IMDB-M ↑ | RE-B ↑ | RE-M5K ↑ | RE-M12K ↑ | Average |
|---|---|---|---|---|---|---|
| GCN | 76.00 ±3.44 | 50.33 ±1.89 | 91.15 ±1.63 | 56.47 ±1.54 | 48.71 ±0.88 | 64.53 |
| GFuN | 68.90 ±3.42 | 51.27 ±3.22 | 92.25 ±1.12 | 57.53 ±1.31 | 51.54 ±1.19 | 64.30 |

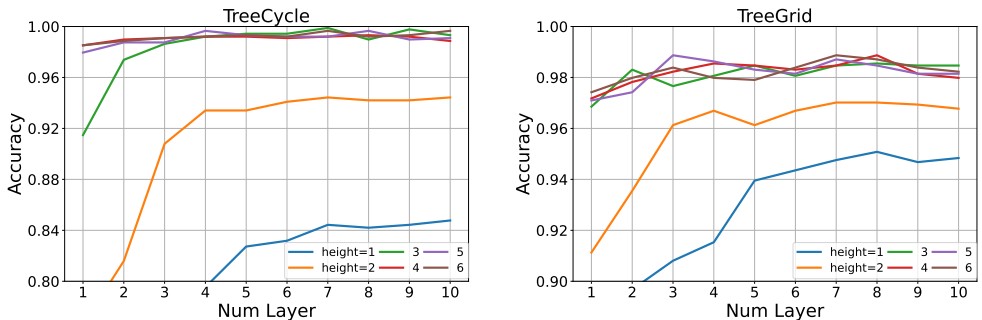

Figure 7: Node Classification accuracy for `MeGraph` model on TreeCycle (left) and TreeGrid (right) datasets, varying the height $h$ and the number of `Mee` layers $n$. A clear gap can be observed between heights 1, 2 and $\geq 3$. The concrete number of accuracy can be found in Table 14.

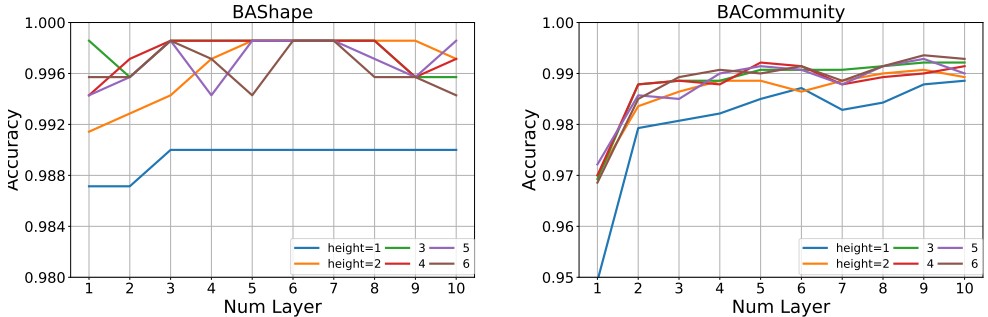

Figure 8: Node Classification accuracy for `MeGraph` model on BAShape (left) and BACommunity (right) datasets, varying the height $h$ and the number of `Mee` layers $n$. A clear gap can be observed between heights 1 and $\geq 2$. The concrete number of accuracy can be found in Table 15.

## F.7 Graph Theory Dataset

We provide a list of tables (from Table 19 to 30) showing the individual results of Table 1 for each possible graph generation method. Each table contains a list of variants of models and 5 tasks. Some graph generation methods and task combinations are trivial so we filter them out.

Table 14: Node Classification accuracy for `MeGraph` model on TreeCycle (above) and TreeGrid (below).

| layer \ height | 1 | 2 | 3 | 4 | 5 | 6 |
|---|---|---|---|---|---|---|
| 1 | 61.48 ±6.04 | 76.59 ±4.41 | 91.48 ±2.70 | 98.52 ±1.69 | 97.95 ±2.32 | 98.52 ±1.35 |
| 2 | 67.27 ±6.91 | 81.59 ±4.03 | 97.39 ±1.25 | 98.98 ±0.94 | 98.75 ±1.29 | 98.86 ±1.14 |
| 3 | 74.43 ±3.60 | 90.80 ±2.61 | 98.64 ±1.11 | 99.09 ±1.11 | 98.75 ±1.56 | 99.09 ±0.85 |
| 4 | 79.55 ±4.34 | 93.41 ±2.82 | 99.20 ±0.73 | 99.20 ±0.89 | 99.66 ±0.73 | 99.20 ±1.69 |
| 5 | 82.73 ±4.06 | 93.41 ±1.89 | 99.43 ±1.05 | 99.20 ±1.35 | 99.32 ±1.16 | 99.32 ±0.56 |
| 6 | 83.18 ±3.51 | 94.09 ±2.02 | 99.43 ±0.76 | 99.09 ±0.85 | 99.20 ±1.69 | 99.20 ±0.89 |
| 7 | 84.43 ±3.74 | 94.43 ±2.24 | 99.89 ±0.34 | 99.20 ±1.02 | 99.20 ±0.89 | 99.66 ±0.73 |
| 8 | 84.20 ±3.82 | 94.20 ±2.00 | 98.98 ±1.19 | 99.32 ±0.75 | 99.66 ±0.52 | 99.20 ±0.73 |
| 9 | 84.43 ±3.87 | 94.20 ±2.06 | 99.77 ±0.45 | 99.20 ±1.02 | 98.98 ±1.07 | 99.32 ±0.75 |
| 10 | 84.77 ±3.98 | 94.43 ±2.18 | 99.32 ±0.75 | 98.86 ±1.14 | 99.09 ±1.67 | 99.66 ±0.52 |

| layer \ height | 1 | 2 | 3 | 4 | 5 | 6 |
|---|---|---|---|---|---|---|
| 1 | 79.11 ±3.07 | 91.13 ±2.01 | 96.85 ±1.11 | 97.18 ±1.31 | 97.10 ±1.45 | 97.42 ±1.34 |
| 2 | 89.68 ±1.76 | 93.55 ±1.53 | 98.31 ±0.76 | 97.82 ±1.14 | 97.42 ±1.24 | 97.98 ±0.74 |
| 3 | 90.81 ±1.36 | 96.13 ±1.48 | 97.66 ±1.22 | 98.23 ±0.94 | 98.87 ±0.82 | 98.39 ±0.62 |
| 4 | 91.53 ±1.04 | 96.69 ±1.05 | 98.06 ±1.03 | 98.55 ±1.01 | 98.63 ±1.08 | 97.98 ±1.15 |
| 5 | 93.95 ±1.58 | 96.13 ±1.76 | 98.47 ±1.17 | 98.47 ±0.92 | 98.31 ±0.84 | 97.90 ±0.65 |
| 6 | 94.35 ±1.25 | 96.69 ±1.46 | 98.06 ±1.03 | 98.31 ±1.05 | 98.15 ±1.20 | 98.39 ±1.20 |
| 7 | 94.76 ±1.10 | 97.02 ±1.44 | 98.47 ±0.84 | 98.47 ±1.05 | 98.71 ±0.74 | 98.87 ±0.90 |
| 8 | 95.08 ±0.76 | 97.02 ±1.20 | 98.55 ±1.24 | 98.87 ±0.82 | 98.47 ±0.92 | 98.71 ±1.15 |
| 9 | 94.68 ±1.09 | 96.94 ±1.19 | 98.47 ±0.43 | 98.15 ±1.20 | 98.15 ±0.89 | 98.39 ±1.08 |
| 10 | 94.84 ±1.21 | 96.77 ±1.20 | 98.47 ±0.92 | 97.98 ±1.50 | 98.15 ±1.02 | 98.23 ±1.19 |

Table 15: Node Classification accuracy for `MeGraph` model on BAShape (above) and BACommunity (below).

| layer \ height | 1 | 2 | 3 | 4 | 5 | 6 |
|---|---|---|---|---|---|---|
| 1 | 98.71 ±1.00 | 99.14 ±1.14 | 99.86 ±0.43 | 99.43 ±0.70 | 99.43 ±0.95 | 99.57 ±0.91 |
| 2 | 98.71 ±1.00 | 99.29 ±0.96 | 99.57 ±0.91 | 99.71 ±0.57 | 99.57 ±0.91 | 99.57 ±0.91 |
| 3 | 99.00 ±0.91 | 99.43 ±0.95 | 99.86 ±0.43 | 99.86 ±0.43 | 99.86 ±0.43 | 99.86 ±0.43 |
| 4 | 99.00 ±0.91 | 99.71 ±0.57 | 99.86 ±0.43 | 99.86 ±0.43 | 99.43 ±0.95 | 99.71 ±0.57 |
| 5 | 99.00 ±0.91 | 99.86 ±0.43 | 99.86 ±0.43 | 99.86 ±0.43 | 99.86 ±0.43 | 99.43 ±0.95 |
| 6 | 99.00 ±0.91 | 99.86 ±0.43 | 99.86 ±0.43 | 99.86 ±0.43 | 99.86 ±0.43 | 99.86 ±0.43 |
| 7 | 99.00 ±0.91 | 99.86 ±0.43 | 99.86 ±0.43 | 99.86 ±0.43 | 99.86 ±0.43 | 99.86 ±0.43 |
| 8 | 99.00 ±0.91 | 99.86 ±0.43 | 99.86 ±0.43 | 99.86 ±0.43 | 99.71 ±0.57 | 99.57 ±0.65 |
| 9 | 99.00 ±0.91 | 99.86 ±0.43 | 99.57 ±0.91 | 99.57 ±0.91 | 99.57 ±0.91 | 99.57 ±0.91 |
| 10 | 99.00 ±0.91 | 99.71 ±0.57 | 99.57 ±0.91 | 99.71 ±0.57 | 99.86 ±0.43 | 99.43 ±0.95 |

| layer \ height | 1 | 2 | 3 | 4 | 5 | 6 |
|---|---|---|---|---|---|---|
| 1 | 94.93 ±1.30 | 97.00 ±1.80 | 96.93 ±1.60 | 97.00 ±1.88 | 97.21 ±1.70 | 96.86 ±1.67 |
| 2 | 97.93 ±0.87 | 98.36 ±0.72 | 98.79 ±0.46 | 98.79 ±0.46 | 98.57 ±0.55 | 98.50 ±1.03 |
| 3 | 98.07 ±0.91 | 98.64 ±0.87 | 98.86 ±0.91 | 98.86 ±0.80 | 98.50 ±0.98 | 98.93 ±0.80 |
| 4 | 98.21 ±0.97 | 98.86 ±0.65 | 98.86 ±0.80 | 98.79 ±0.64 | 99.00 ±0.73 | 99.07 ±0.64 |
| 5 | 98.50 ±0.87 | 98.86 ±0.91 | 99.07 ±0.64 | 99.21 ±0.67 | 99.14 ±0.70 | 99.00 ±0.73 |
| 6 | 98.71 ±0.83 | 98.64 ±0.87 | 99.07 ±0.64 | 99.14 ±0.70 | 99.07 ±0.85 | 99.14 ±0.70 |
| 7 | 98.29 ±0.91 | 98.86 ±0.65 | 99.07 ±0.56 | 98.79 ±0.56 | 98.79 ±0.72 | 98.86 ±0.57 |
| 8 | 98.43 ±0.77 | 99.00 ±0.47 | 99.14 ±0.53 | 98.93 ±0.58 | 99.14 ±0.29 | 99.14 ±0.43 |
| 9 | 98.79 ±0.79 | 99.07 ±0.56 | 99.21 ±0.50 | 99.00 ±0.73 | 99.29 ±0.45 | 99.36 ±0.50 |
| 10 | 98.86 ±0.73 | 98.93 ±0.80 | 99.21 ±0.87 | 99.14 ±0.70 | 99.00 ±0.73 | 99.29 ±0.64 |

Table 16: Comparison results of `MeGraph` with ATT and GATE on Graph Theory Benchmark.

| Category | Model | $SP_{sssd}$ | MCC | Diameter | $SP_{ss}$ | ECC |
|---|---|---|---|---|---|---|
| MeGraph w. ATT | $h=1$ | 2.990 ±3.411 | 3.346 ±3.228 | 44.41 ±36.33 | 16.39 ±13.57 | 29.04 ±27.59 |
| | $h=5$ | 0.594 ±0.903 | 1.706 ±1.409 | 3.256 ±2.956 | 1.018 ±1.071 | 14.80 ±17.09 |
| | $h=5, \eta_v=0.3, \tau_c=4$ | 0.749 ±1.131 | 1.128 ±0.794 | 4.430 ±4.329 | 0.640 ±0.833 | 5.649 ±4.496 |
| MeGraph w. GATE | $h=1$ | 4.144 ±4.181 | 0.908 ±0.934 | 6.343 ±7.152 | 13.94 ±12.78 | 19.73 ±19.47 |
| | $h=5$ | 0.809 ±0.993 | 0.660 ±0.601 | 2.506 ±2.639 | 0.669 ±0.546 | 7.508 ±7.558 |
| | $h=5, \eta_v=0.3, \tau_c=4$ | 0.602 ±0.622 | 0.599 ±0.520 | 0.544 ±0.490 | 0.342 ±0.193 | 0.859 ±0.712 |

Table 17: Comparison results of `MeGraph` with ATT and GATE on GNN Benchmark.

| | ZINC | AQSOL | CIFAR10 | MNIST | PATTERN | CLUSTER |
|---|---|---|---|---|---|---|
| MeGraph w. ATT ($h=1$) | 0.4258 ±0.0054 | 1.1421 ±0.0270 | 69.890 ±0.209 | 97.570 ±0.168 | 78.232 ±0.827 | 59.497 ±0.207 |
| MeGraph w. ATT ($h=5$) | 0.3637 ±0.0116 | 1.0767 ±0.0105 | 69.925 ±0.631 | 97.860 ±0.098 | 83.798 ±0.885 | 68.930 ±68.76 |
| MeGraph w. GATE ($h=1$) | 0.3336 ±0.0036 | 1.0766 ±1.0556 | 64.200 ±0.586 | 96.812 ±0.205 | 85.391 ±0.029 | 59.321 ±0.290 |
| MeGraph w. GATE ($h=5$) | 0.2897 ±0.0291 | 1.0240 ±0.0098 | 64.935 ±0.829 | 97.290 ±0.140 | 86.611 ±0.041 | 67.122 ±3.323 |

Table 18: Comparison results of `MeGraph` with ATT and GATE on OGB-G.

| Model | molhiv ↑ | molbace ↑ | molbbbp ↑ | molclintox ↑ | molsider ↑ |
|---|---|---|---|---|---|
| MeGraph w. ATT ($h=1$) | 77.33 ±0.78 | 74.83 ±4.87 | 64.74 ±1.14 | 86.34 ±1.04 | 58.12 ±0.53 |
| MeGraph w. ATT ($h=5$) | 77.15 ±1.37 | 76.13 ±3.85 | 68.68 ±2.07 | 87.17 ±0.76 | 58.03 ±1.58 |
| MeGraph w. GATE ($h=1$) | 76.35 ±0.70 | 76.36 ±1.55 | 65.97 ±1.98 | 87.12 ±0.74 | 58.11 ±1.29 |
| MeGraph w. GATE ($h=5$) | 78.14 ±0.91 | 78.90 ±1.29 | 66.53 ±0.74 | 89.02 ±2.56 | 59.58 ±1.88 |

| Model | moltox21 ↑ | moltoxcast ↑ | molesol ↓ | molfreesolv ↓ | mollipo ↓ |
|---|---|---|---|---|---|
| MeGraph w. ATT ($h=1$) | 75.88 ±0.64 | 64.65 ±0.59 | 1.091 ±0.030 | 2.318 ±0.089 | 0.790 ±0.012 |
| MeGraph w. ATT ($h=5$) | 76.71 ±0.98 | 66.98 ±0.70 | 1.007 ±0.617 | 2.065 ±0.151 | 0.736 ±0.023 |
| MeGraph w. GATE ($h=1$) | 75.30 ±0.43 | 64.34 ±0.62 | 1.064 ±0.015 | 2.191 ±0.068 | 0.766 ±0.008 |
| MeGraph w. GATE ($h=5$) | 76.91 ±0.25 | 66.78 ±0.13 | 1.003 ±0.086 | 2.048 ±0.199 | 0.700 ±0.014 |

Table 19: Graph Theory Benchmark results on Grid graphs, all results are obtained using our codebase.

| Category | Model | $SP_{sssd}$ | MCC | Diameter | $SP_{ss}$ | ECC |
|---|---|---|---|---|---|---|
| Baselines ($h=1$) | $n=1$ | 6.60 ±0.541 | 1.50 ±0.050 | 22.49 ±1.36 | 26.74 ±0.347 | 20.99 ±0.232 |
| | $n=5$ | 4.18 ±0.737 | 1.29 ±0.124 | 5.04 ±1.26 | 15.54 ±0.155 | 20.32 ±0.326 |
| | $n=10$ | 3.70 ±0.422 | 1.33 ±0.100 | 0.737 ±0.116 | 7.24 ±0.243 | 20.32 ±0.422 |
| MeGraph($h=5$) EdgePool($\tau_c=2$) | $n=1$ | 1.19 ±0.486 | 1.24 ±0.154 | 6.78 ±1.95 | 5.34 ±0.265 | 18.00 ±0.910 |
| | $n=5$ | 0.738 ±0.322 | 1.11 ±0.043 | 0.616 ±0.310 | 0.617 ±0.099 | 13.3 ±3.31 |
| MeGraph S-EdgePool Variants ($h=5, n=5$) | $\tau_c=3$ | 0.361 ±0.182 | 1.24 ±0.113 | 0.382 ±0.120 | 0.442 ±0.130 | 0.918 ±0.220 |
| | $\eta_v=0.3$ | 4.77 ±2.50 | 1.33 ±0.161 | 0.349 ±0.074 | 5.40 ±0.954 | 3.59 ±0.354 |
| | $\eta_v=0.3, \tau_c=4$ | 0.745 ±0.316 | 1.35 ±0.168 | 0.385 ±0.168 | 0.552 ±0.113 | 0.622 ±0.100 |
| | $\eta_v=0.5, \tau_c=4$ | 1.61 ±0.394 | 1.28 ±0.138 | 0.458 ±0.220 | 1.71 ±0.535 | 1.48 ±0.283 |
| | $\eta_v=0.3, \tau_c=4$ (X-Pool) | 1.03 ±0.365 | 1.50 ±0.142 | 0.626 ±0.216 | 1.70 ±0.185 | 3.44 ±0.991 |
| | $\eta_v=0.3, \tau_c=4$ (w/o pw) | 0.616 ±0.194 | 1.66 ±0.083 | 0.361 ±0.147 | 0.678 ±0.139 | 1.70 ±0.485 |
| Graph-UNets | $h=5,n=9,\eta_v=0.3,\tau_c=4$ | 0.969 ±0.643 | 2.18 ±0.381 | 0.111 ±0.091 | 0.773 ±0.086 | 0.548 ±0.131 |

Table 20: Graph Theory Benchmark results on Tree graphs. All results are obtained using our codebase.

| Category | Model | $SP_{sssd}$ | MCC | Diameter | $SP_{ss}$ | ECC |
|---|---|---|---|---|---|---|
| Baselines ($h=1$) | $n=1$ | 5.21 ±0.209 | 1.28 ±0.050 | 3.77 ±1.22 | 17.16 ±0.168 | 24.63 ±0.427 |
| | $n=5$ | 3.34 ±0.375 | 0.405 ±0.089 | 0.504 ±0.109 | 7.66 ±0.325 | 18.11 ±1.85 |
| | $n=10$ | 3.16 ±0.252 | 0.338 ±0.046 | 0.100 ±0.059 | 2.28 ±0.209 | 14.93 ±0.800 |
| MeGraph($h=5$) EdgePool($\tau_c=2$) | $n=1$ | 1.62 ±0.314 | 0.846 ±0.071 | 0.725 ±0.249 | 6.99 ±0.610 | 12.27 ±0.843 |
| | $n=5$ | 0.83 ±0.667 | 0.490 ±0.118 | 0.084 ±0.030 | 1.27 ±0.442 | 2.87 ±0.420 |
| MeGraph S-EdgePool Variants ($h=5, n=5$) | $\tau_c=3$ | 0.599 ±0.200 | 0.483 ±0.081 | 0.075 ±0.012 | 0.497 ±0.121 | 0.429 ±0.105 |
| | $\eta_v=0.3$ | 0.868 ±0.230 | 0.413 ±0.054 | 0.142 ±0.047 | 0.789 ±0.092 | 0.534 ±0.074 |
| | $\eta_v=0.3, \tau_c=4$ | 0.615 ±0.209 | 0.418 ±0.024 | 0.081 ±0.017 | 0.440 ±0.106 | 0.436 ±0.097 |
| | $\eta_v=0.5, \tau_c=4$ | 1.06 ±0.327 | 0.424 ±0.042 | 0.214 ±0.018 | 1.20 ±0.128 | 2.03 ±0.507 |
| | $\eta_v=0.3, \tau_c=4$ (X-Pool) | 0.666 ±0.118 | 0.596 ±0.067 | 0.182 ±0.057 | 1.22 ±0.281 | 1.11 ±0.122 |
| | $\eta_v=0.3, \tau_c=4$ (w/o pw) | 0.771 ±0.141 | 0.455 ±0.056 | 0.124 ±0.032 | 0.700 ±0.190 | 1.07 ±0.260 |
| Graph-UNets | $h=5,n=9,\eta_v=0.3,\tau_c=4$ | 0.873 ±0.247 | 0.667 ±0.043 | 0.804 ±0.284 | 0.606 ±0.123 | 1.00 ±0.221 |

Table 21: Graph Theory Benchmark results on Ladder graphs, all results are obtained using our codebase.

| Category | Model | $SP_{sssd}$ | MCC | Diameter | $SP_{ss}$ | ECC |
|---|---|---|---|---|---|---|
| Baselines ($h$=1) | $n$=1 | 5.06 ±0.330 | 1.73 ±0.249 | 1.17 ±0.149 | 13.20 ±0.126 | 20.10 ±0.583 |
| | $n$=5 | 0.692 ±0.204 | 0.734 ±0.106 | 1.39 ±0.078 | 5.02 ±0.876 | 19.81 ±0.669 |
| | $n$=10 | 0.257 ±0.078 | 0.691 ±0.119 | 1.55 ±0.069 | 1.60 ±0.194 | 20.40 ±0.995 |
| MeGraph($h$=5) | $n$=1 | 0.662 ±0.165 | 0.866 ±0.071 | 1.57 ±0.992 | 2.18 ±0.181 | 6.61 ±1.32 |
| EdgePool($\tau_c$=2) | $n$=5 | 0.251 ±0.108 | 0.753 ±0.091 | 0.175 ±0.169 | 0.321 ±0.058 | 1.18 ±0.746 |
| MeGraph S-EdgePool Variants ($h$=5, $n$=5) | $\tau_c$=3 | 0.296 ±0.070 | 0.754 ±0.086 | 0.226 ±0.069 | 0.228 ±0.021 | 0.285 ±0.069 |
| | $\eta_v$=0.3 | 0.507 ±0.204 | 0.768 ±0.050 | 0.156 ±0.053 | 0.969 ±0.148 | 0.787 ±0.059 |
| | $\eta_v$=0.3, $\tau_c$=4 | 0.297 ±0.113 | 0.712 ±0.059 | 0.095 ±0.046 | 0.180 ±0.026 | 0.225 ±0.043 |
| | $\eta_v$=0.5, $\tau_c$=4 | 0.375 ±0.196 | 0.656 ±0.064 | 0.058 ±0.019 | 0.612 ±0.191 | 0.464 ±0.121 |
| | $\eta_v$=0.3, $\tau_c$=4 (X-Pool) | 0.442 ±0.108 | 0.742 ±0.047 | 0.158 ±0.074 | 0.710 ±0.076 | 0.765 ±0.089 |
| | $\eta_v$=0.3, $\tau_c$=4 (w/o pw) | 0.245 ±0.021 | 0.682 ±0.105 | 0.106 ±0.052 | 0.271 ±0.039 | 0.618 ±0.188 |
| Graph-UNets | $h$=5,$n$=9,$\eta_v$=0.3,$\tau_c$=4 | 0.339 ±0.11 | 0.797 ±0.138 | 0.013 ±0.005 | 0.287 ±0.023 | 0.230 ±0.054 |

Table 22: Graph Theory Benchmark results on Line graphs, all results are obtained using our codebase.

| Category | Model | $SP_{sssd}$ | MCC | Diameter | $SP_{ss}$ | ECC |
|---|---|---|---|---|---|---|
| Baselines ($h$=1) | $n$=1 | 30.37 ±1.41 | 0.458 ±0.035 | 21.49 ±8.84 | 68.99 ±0.247 | 75.46 ±1.86 |
| | $n$=5 | 10.55 ±2.40 | 0.019 ±0.004 | 9.97 ±10.85 | 46.39 ±3.09 | 78.49 ±4.38 |
| | $n$=10 | 3.29 ±0.813 | 0.012 ±0.003 | 10.18 ±10.59 | 35.07 ±2.71 | 77.23 ±3.42 |
| MeGraph($h$=5) | $n$=1 | 1.45 ±0.598 | 0.056 ±0.014 | 7.62 ±4.43 | 10.13 ±2.33 | 45.19 ±8.64 |
| EdgePool($\tau_c$=2) | $n$=5 | 0.536 ±0.149 | 0.016 ±0.007 | 0.611 ±0.238 | 1.06 ±0.341 | 14.12 ±13.82 |
| MeGraph S-EdgePool Variants ($h$=5, $n$=5) | $\tau_c$=3 | 0.349 ±0.206 | 0.013 ±0.003 | 0.724 ±0.479 | 0.339 ±0.102 | 1.15 ±0.267 |
| | $\eta_v$=0.3 | 3.65 ±2.13 | 0.017 ±0.005 | 1.75 ±1.63 | 13.99 ±2.09 | 7.45 ±0.989 |
| | $\eta_v$=0.3, $\tau_c$=4 | 0.283 ±0.072 | 0.019 ±0.006 | 0.584 ±0.337 | 0.515 ±0.044 | 1.27 ±1.08 |
| | $\eta_v$=0.5, $\tau_c$=4 | 1.81 ±0.121 | 0.022 ±0.006 | 0.711 ±0.213 | 2.64 ±0.047 | 3.77 ±0.763 |
| | $\eta_v$=0.3, $\tau_c$=4 (X-Pool) | 1.06 ±0.510 | 0.101 ±0.016 | 0.767 ±0.522 | 2.29 ±0.472 | 3.89 ±1.02 |
| | $\eta_v$=0.3, $\tau_c$=4 (w/o pw) | 0.377 ±0.106 | 0.022 ±0.007 | 1.19 ±1.17 | 1.12 ±0.115 | 3.34 ±0.904 |
| Graph-UNets | $h$=5,$n$=9,$\eta_v$=0.3,$\tau_c$=4 | 0.426 ±0.223 | 0.062 ±0.008 | 2.89 ±1.89 | 0.767 ±0.129 | 4.78 ±1.94 |

Table 23: Graph Theory Benchmark results on Caterpillar graphs, all results are obtained using our codebase.

| Category | Model | $SP_{sssd}$ | MCC | Diameter | $SP_{ss}$ | ECC |
|---|---|---|---|---|---|---|
| Baselines ($h$=1) | $n$=1 | 24.24 ±1.57 | 1.25 ±0.082 | 28.62 ±2.55 | 19.08 ±0.208 | 35.32 ±0.462 |
| | $n$=5 | 8.32 ±2.10 | 0.561 ±0.070 | 4.59 ±0.346 | 9.62 ±0.357 | 37.01 ±1.48 |
| | $n$=10 | 6.40 ±0.652 | 0.630 ±0.127 | 5.06 ±0.499 | 4.06 ±0.297 | 37.87 ±3.22 |
| MeGraph($h$=5) | $n$=1 | 5.04 ±1.03 | 0.685 ±0.077 | 6.08 ±1.40 | 5.40 ±0.843 | 28.52 ±2.16 |
| EdgePool($\tau_c$=2) | $n$=5 | 3.44 ±1.13 | 0.533 ±0.064 | 2.00 ±1.28 | 0.921 ±0.149 | 5.20 ±1.57 |
| MeGraph S-EdgePool Variants ($h$=5, $n$=5) | $\tau_c$=3 | 2.47 ±0.529 | 0.607 ±0.081 | 0.591 ±0.172 | 0.574 ±0.073 | 1.21 ±0.148 |
| | $\eta_v$=0.3 | 3.61 ±1.36 | 0.582 ±0.052 | 0.578 ±0.231 | 1.69 ±0.572 | 1.95 ±0.322 |
| | $\eta_v$=0.3, $\tau_c$=4 | 1.59 ±0.444 | 0.535 ±0.091 | 0.317 ±0.104 | 0.474 ±0.170 | 1.32 ±0.272 |
| | $\eta_v$=0.5, $\tau_c$=4 | 2.00 ±0.648 | 0.514 ±0.040 | 1.10 ±0.288 | 0.986 ±0.130 | 2.11 ±0.766 |
| | $\eta_v$=0.3, $\tau_c$=4 (X-Pool) | 1.39 ±0.478 | 0.602 ±0.110 | 0.736 ±0.230 | 1.78 ±0.254 | 3.36 ±0.873 |
| | $\eta_v$=0.3, $\tau_c$=4 (w/o pw) | 1.82 ±0.627 | 0.628 ±0.093 | 0.604 ±0.067 | 0.797 ±0.299 | 2.25 ±0.230 |
| Graph-UNets | $h$=5,$n$=9,$\eta_v$=0.3,$\tau_c$=4 | 1.57 ±0.670 | 0.679 ±0.098 | 3.18 ±0.583 | 0.976 ±0.270 | 3.83 ±1.06 |

Table 24: Graph Theory Benchmark results on Lobster graphs, all results are obtained using our codebase.

| Category | Model | $SP_{sssd}$ | MCC | Diameter | $SP_{ss}$ | ECC |
|---|---|---|---|---|---|---|
| Baselines (h=1) | n=1 | 23.92 ±0.319 | 1.06 ±0.166 | 11.93 ±1.32 | 38.44 ±0.065 | 40.46 ±0.350 |
| | n=5 | 10.89 ±1.47 | 0.544 ±0.067 | 3.66 ±0.424 | 20.12 ±0.105 | 28.81 ±1.14 |
| | n=10 | 7.35 ±2.50 | 0.631 ±0.067 | 2.59 ±0.517 | 10.52 ±0.619 | 28.47 ±1.65 |
| MeGraph(h=5) EdgePool($\tau_c$=2) | n=1 | 6.00 ±1.82 | 0.785 ±0.062 | 4.35 ±1.51 | 13.75 ±0.675 | 30.49 ±2.18 |
| | n=5 | 1.93 ±0.861 | 0.543 ±0.073 | 1.07 ±0.114 | 2.05 ±0.393 | 11.39 ±5.43 |
| MeGraph S-EdgePool Variants (h=5, n=5) | $\tau_c$=3 | 2.02 ±0.791 | 0.447 ±0.123 | 0.705 ±0.133 | 1.66 ±0.270 | 2.23 ±0.378 |
| | $\eta_v$=0.3 | 6.01 ±1.52 | 0.521 ±0.028 | 0.707 ±0.202 | 3.04 ±0.250 | 2.70 ±0.212 |
| | $\eta_v$=0.3, $\tau_c$=4 | 1.90 ±0.449 | 0.489 ±0.069 | 0.671 ±0.165 | 1.30 ±0.106 | 2.62 ±0.849 |
| | $\eta_v$=0.5, $\tau_c$=4 | 3.27 ±0.716 | 0.451 ±0.090 | 0.941 ±0.324 | 2.82 ±0.803 | 4.04 ±0.527 |
| | $\eta_v$=0.3, $\tau_c$=4 (X-Pool) | 2.67 ±0.486 | 0.494 ±0.109 | 1.01 ±0.194 | 2.79 ±0.343 | 4.16 ±0.886 |
| | $\eta_v$=0.3, $\tau_c$=4 (w/o pw) | 1.85 ±0.432 | 0.473 ±0.069 | 0.892 ±0.277 | 1.77 ±0.329 | 4.33 ±1.71 |
| Graph-UNets | h=5,n=9,$\eta_v$=0.3,$\tau_c$=4 | 4.85 ±1.48 | 0.782 ±0.026 | 3.74 ±0.361 | 2.96 ±0.443 | 4.25 ±0.544 |

Table 25: Graph Theory Benchmark results on Cycle graphs, all results are obtained using our codebase.

| Category | Model | $SP_{sssd}$ | MCC | Diameter | $SP_{ss}$ | ECC |
|---|---|---|---|---|---|---|
| Baselines (h=1) | n=1 | 18.75 ±0.066 | 0.534 ±0.022 | 22.35 ±0.149 | 24.07 ±0.009 | 21.47 ±0.060 |
| | n=5 | 3.39 ±0.304 | 0.027 ±0.001 | 25.11 ±0.325 | 12.44 ±1.05 | 21.81 ±0.102 |
| | n=10 | 0.352 ±0.060 | 0.011 ±0.003 | 26.54 ±1.16 | 8.65 ±1.02 | 24.09 ±0.360 |
| MeGraph(h=5) EdgePool($\tau_c$=2) | n=1 | 0.594 ±0.212 | 0.074 ±0.029 | 9.11 ±1.88 | 4.07 ±0.364 | 21.53 ±0.070 |
| | n=5 | 0.060 ±0.032 | 0.014 ±0.003 | 13.44 ±6.40 | 0.103 ±0.016 | 24.05 ±0.204 |
| MeGraph S-EdgePool Variants (h=5, n=5) | $\tau_c$=3 | 0.066 ±0.036 | 0.015 ±0.006 | 0.241 ±0.049 | 0.090 ±0.037 | 0.342 ±0.186 |
| | $\eta_v$=0.3 | 2.45 ±0.873 | 0.015 ±0.001 | 0.709 ±0.226 | 8.36 ±0.261 | 0.488 ±0.267 |
| | $\eta_v$=0.3, $\tau_c$=4 | 0.060 ±0.030 | 0.019 ±0.003 | 0.312 ±0.236 | 0.226 ±0.050 | 0.562 ±0.209 |
| | $\eta_v$=0.5, $\tau_c$=4 | 0.451 ±0.203 | 0.014 ±0.004 | 0.252 ±0.124 | 1.05 ±0.524 | 4.30 ±1.90 |
| | $\eta_v$=0.3, $\tau_c$=4 (X-Pool) | 0.494 ±0.292 | 0.096 ±0.028 | 0.468 ±0.220 | 1.08 ±0.130 | 0.860 ±0.292 |
| | $\eta_v$=0.3, $\tau_c$=4 (w/o pw) | 0.159 ±0.209 | 0.017 ±0.008 | 1.23 ±0.928 | 0.461 ±0.118 | 8.26 ±3.70 |
| Graph-UNets | h=5,n=9,$\eta_v$=0.3,$\tau_c$=4 | 0.144 ±0.073 | 0.035 ±0.010 | 3.21 ±0.893 | 0.439 ±0.089 | 5.91 ±1.31 |

Table 26: Graph Theory Benchmark results on Pseudotree graphs, all results are obtained using our codebase.

| Category | Model | $SP_{sssd}$ | MCC | Diameter | $SP_{ss}$ | ECC |
|---|---|---|---|---|---|---|
| Baselines (h=1) | n=1 | 1.93 ±0.239 | 1.71 ±0.281 | 2.78 ±0.098 | 6.27 ±0.004 | 4.23 ±0.034 |
| | n=5 | 0.061 ±0.024 | 0.942 ±0.094 | 1.74 ±0.299 | 1.54 ±0.006 | 4.15 ±0.086 |
| | n=10 | 0.037 ±0.022 | 0.775 ±0.094 | 1.84 ±0.260 | 0.126 ±0.038 | 4.06 ±0.037 |
| MeGraph(h=5) EdgePool($\tau_c$=2) | n=1 | 0.404 ±0.096 | 1.75 ±0.133 | 1.50 ±0.494 | 2.25 ±0.280 | 3.97 ±0.270 |
| | n=5 | 0.141 ±0.022 | 0.999 ±0.054 | 1.16 ±0.069 | 0.148 ±0.034 | 3.12 ±0.202 |
| MeGraph S-EdgePool Variants (h=5, n=5) | $\tau_c$=3 | 0.130 ±0.069 | 0.912 ±0.073 | 0.669 ±0.080 | 0.115 ±0.015 | 0.797 ±0.079 |
| | $\eta_v$=0.3 | 0.048 ±0.030 | 0.839 ±0.077 | 0.758 ±0.134 | 0.246 ±0.021 | 0.838 ±0.023 |
| | $\eta_v$=0.3, $\tau_c$=4 | 0.106 ±0.054 | 0.814 ±0.092 | 0.663 ±0.076 | 0.133 ±0.028 | 0.845 ±0.101 |
| | $\eta_v$=0.5, $\tau_c$=4 | 0.071 ±0.048 | 1.03 ±0.186 | 0.583 ±0.065 | 0.171 ±0.038 | 0.868 ±0.034 |
| | $\eta_v$=0.3, $\tau_c$=4 (X-Pool) | 0.564 ±0.155 | 0.966 ±0.172 | 0.977 ±0.054 | 0.611 ±0.065 | 1.10 ±0.036 |
| | $\eta_v$=0.3, $\tau_c$=4 (w/o pw) | 0.080 ±0.033 | 0.971 ±0.072 | 0.956 ±0.230 | 0.276 ±0.017 | 1.13 ±0.321 |
| Graph-UNets | h=5,n=9,$\eta_v$=0.3,$\tau_c$=4 | 0.467 ±0.065 | 1.09 ±0.072 | 1.71 ±0.295 | 0.721 ±0.092 | 2.25 ±0.327 |

Table 27: Graph Theory Benchmark results on Geo graphs, all results are obtained using our codebase.

| Category | Model | $SP_{sssd}$ | MCC | Diameter | $SP_{ss}$ | ECC |
|---|---|---|---|---|---|---|
| Baselines ($h$=1) | $n$=1 | 5.79 ±0.630 | 0.424 ±0.023 | 11.85 ±0.391 | 12.49 ±0.035 | 14.82 ±0.056 |
| | $n$=5 | 1.02 ±0.772 | 0.407 ±0.040 | 8.37 ±0.468 | 5.10 ±0.435 | 14.33 ±0.079 |
| | $n$=10 | 0.304 ±0.125 | 0.404 ±0.061 | 9.41 ±0.759 | 0.803 ±0.162 | 14.33 ±0.136 |
| MeGraph($h$=5) EdgePool($\tau_c$=2) | $n$=1 | 1.60 ±0.880 | 0.347 ±0.033 | 10.17 ±2.04 | 4.87 ±0.777 | 11.91 ±0.451 |
| | $n$=5 | 0.232 ±0.061 | 0.273 ±0.018 | 2.70 ±0.288 | 0.575 ±0.127 | 6.92 ±2.36 |
| MeGraph S-EdgePool Variants ($h$=5, $n$=5) | $\tau_c$=3 | 0.188 ±0.100 | 0.288 ±0.020 | 2.04 ±0.225 | 0.562 ±0.186 | 2.42 ±0.333 |
| | $\eta_v$=0.3 | 1.38 ±0.617 | 0.330 ±0.025 | 4.40 ±1.15 | 1.37 ±0.083 | 5.45 ±0.465 |
| | $\eta_v$=0.3, $\tau_c$=4 | 0.230 ±0.070 | 0.231 ±0.034 | 1.99 ±0.549 | 0.454 ±0.057 | 2.69 ±0.369 |
| | $\eta_v$=0.5, $\tau_c$=4 | 0.374 ±0.148 | 0.368 ±0.043 | 3.95 ±0.319 | 0.777 ±0.122 | 4.61 ±0.717 |
| | $\eta_v$=0.3, $\tau_c$=4 (X-Pool) | 1.04 ±0.502 | 0.362 ±0.031 | 2.32 ±0.440 | 2.37 ±0.260 | 5.08 ±0.737 |
| | $\eta_v$=0.3, $\tau_c$=4 (w/o pw) | 0.233 ±0.046 | 0.261 ±0.035 | 2.58 ±0.617 | 1.09 ±0.226 | 4.85 ±0.805 |
| Graph-UNets | $h$=5,$n$=9,$\eta_v$=0.3,$\tau_c$=4 | 1.49 ±0.451 | 0.400 ±0.020 | 4.63 ±0.647 | 2.42 ±0.458 | 7.36 ±1.62 |

Table 28: Graph Theory Benchmark results on SBM graphs, all results are obtained using our codebase.

| Category | Model | $SP_{sssd}$ | MCC | Diameter | $SP_{ss}$ | ECC |
|---|---|---|---|---|---|---|
| Baselines ($h$=1) | $n$=1 | 1.14 ±0.084 | 3.28 ±0.135 | 3.05 ±0.171 | 1.43 ±0.020 | 3.47 ±0.026 |
| | $n$=5 | 0.420 ±0.018 | 3.14 ±0.131 | 2.77 ±0.383 | 0.100 ±0.050 | 3.18 ±0.093 |
| | $n$=10 | 0.704 ±0.264 | 3.29 ±0.063 | 2.70 ±0.235 | 0.012 ±0.005 | 2.97 ±0.051 |
| MeGraph($h$=5) EdgePool($\tau_c$=2) | $n$=1 | 0.786 ±0.145 | 2.79 ±0.226 | 2.49 ±0.692 | 0.547 ±0.059 | 3.35 ±0.195 |
| | $n$=5 | 0.525 ±0.136 | 2.88 ±0.300 | 2.37 ±0.336 | 0.058 ±0.031 | 3.01 ±1.07 |
| MeGraph S-EdgePool Variants ($h$=5, $n$=5) | $\tau_c$=3 | 0.783 ±0.149 | 2.80 ±0.172 | 2.16 ±0.449 | 0.076 ±0.044 | 1.98 ±0.497 |
| | $\eta_v$=0.3 | 1.16 ±0.131 | 3.51 ±0.312 | 2.03 ±0.429 | 0.058 ±0.027 | 1.86 ±0.098 |
| | $\eta_v$=0.3, $\tau_c$=4 | 0.926 ±0.087 | 2.62 ±0.179 | 1.99 ±0.290 | 0.062 ±0.027 | 1.57 ±0.205 |
| | $\eta_v$=0.5, $\tau_c$=4 | 0.798 ±0.255 | 3.14 ±0.234 | 2.05 ±0.375 | 0.063 ±0.013 | 1.70 ±0.049 |
| | $\eta_v$=0.3, $\tau_c$=4 (X-Pool) | 0.935 ±0.106 | 2.60 ±0.374 | 2.16 ±0.204 | 0.056 ±0.014 | 1.87 ±0.111 |
| | $\eta_v$=0.3, $\tau_c$=4 (w/o pw) | 0.788 ±0.063 | 2.52 ±0.100 | 1.38 ±0.131 | 0.484 ±0.063 | 2.21 ±0.169 |
| Graph-UNets | $h$=5,$n$=9,$\eta_v$=0.3,$\tau_c$=4 | 1.51 ±0.138 | 3.57 ±0.569 | 1.96 ±0.240 | 1.71 ±0.115 | 2.77 ±0.160 |

Table 29: Graph Theory Benchmark results on BA graphs, all results are obtained using our codebase.

| Category | Model | $SP_{sssd}$ | MCC | Diameter | $SP_{ss}$ | ECC |
|---|---|---|---|---|---|---|
| Baselines ($h$=1) | $n$=1 | 0.004 ±0.001 | 2.81 ±0.142 | 0.092 ±0.021 | − | 0.128 ±0.006 |
| | $n$=5 | 0.007 ±0.002 | 3.65 ±0.660 | 0.098 ±0.014 | − | 0.091 ±0.011 |
| | $n$=10 | 0.011 ±0.006 | 3.72 ±0.376 | 0.122 ±0.038 | − | 0.080 ±0.004 |
| MeGraph($h$=5) EdgePool($\tau_c$=2) | $n$=1 | 0.006 ±0.004 | 2.00 ±0.380 | 0.101 ±0.020 | − | 0.084 ±0.017 |
| | $n$=5 | 0.003 ±0.001 | 2.00 ±0.240 | 0.104 ±0.011 | − | 0.052 ±0.010 |
| MeGraph S-EdgePool Variants ($h$=5, $n$=5) | $\tau_c$=3 | 0.007 ±0.003 | 1.77 ±0.403 | 0.089 ±0.008 | − | 0.126 ±0.027 |
| | $\eta_v$=0.3 | 0.013 ±0.004 | 1.67 ±0.333 | 0.084 ±0.008 | − | 0.086 ±0.005 |
| | $\eta_v$=0.3, $\tau_c$=4 | 0.011 ±0.005 | 1.42 ±0.252 | 0.073 ±0.015 | − | 0.163 ±0.007 |
| | $\eta_v$=0.5, $\tau_c$=4 | 0.008 ±0.004 | 1.71 ±0.403 | 0.074 ±0.009 | − | 0.156 ±0.021 |
| | $\eta_v$=0.3, $\tau_c$=4 (X-Pool) | 0.009 ±0.003 | 1.22 ±0.242 | 0.088 ±0.021 | − | 0.076 ±0.006 |
| | $\eta_v$=0.3, $\tau_c$=4 (w/o pw) | 0.009 ±0.003 | 1.42 ±0.209 | 0.068 ±0.017 | − | 0.068 ±0.017 |
| Graph-UNets | $h$=5,$n$=9,$\eta_v$=0.3,$\tau_c$=4 | 0.024 ±0.009 | 2.84 ±0.777 | 0.091 ±0.01 | − | 0.179 ±0.0227 |

Table 30: Graph Theory Benchmark results on mixed, ER, Caveman and Star graphs, all results are obtained using our codebase.

| Category | Model | MCC | | | | ECC | |
|---|---|---|---|---|---|---|---|
| | | mix | ER | Caveman | Star | mix | ER |
| Baselines (h=1) | $n$=1 | 3.46 ±0.211 | 2.91 ±0.206 | 0.015 ±0.004 | 0.144 ±0.031 | 0.316 ±0.003 | 0.346 ±0.006 |
| | $n$=5 | 3.29 ±0.261 | 3.35 ±0.205 | 0.014 ±0.003 | 0.078 ±0.021 | 0.228 ±0.008 | 0.289 ±0.008 |
| | $n$=10 | 3.51 ±0.323 | 3.53 ±0.375 | 0.018 ±0.006 | 0.065 ±0.005 | 0.212 ±0.008 | 0.414 ±0.102 |
| MeGraph($h$=5) | $n$=1 | 1.25 ±0.167 | 0.749 ±0.058 | 0.018 ±0.005 | 0.135 ±0.055 | 0.150 ±0.011 | 0.320 ±0.071 |
| EdgePool($\tau_c$=2) | $n$=5 | 1.11 ±0.143 | 0.723 ±0.073 | 0.017 ±0.005 | 0.052 ±0.017 | 0.125 ±0.010 | 0.345 ±0.064 |
| MeGraph S-EdgePool Variants ($h$=5, $n$=5) | $\tau_c$=3 | 1.07 ±0.034 | 0.714 ±0.039 | 0.017 ±0.002 | 0.072 ±0.016 | 0.137 ±0.013 | 0.232 ±0.035 |
| | $\eta_v$=0.3 | 0.908 ±0.153 | 0.627 ±0.090 | 0.026 ±0.007 | 0.125 ±0.026 | 0.128 ±0.014 | 0.248 ±0.012 |
| | $\eta_v$=0.3, $\tau_c$=4 | 1.10 ±0.085 | 0.709 ±0.092 | 0.019 ±0.004 | 0.073 ±0.012 | 0.129 ±0.009 | 0.224 ±0.053 |
| | $\eta_v$=0.5, $\tau_c$=4 | 1.12 ±0.219 | 0.722 ±0.128 | 0.026 ±0.008 | 0.058 ±0.010 | 0.147 ±0.017 | 0.219 ±0.042 |
| | $\eta_v$=0.3, $\tau_c$=4 (X-Pool) | 1.01 ±0.166 | 0.838 ±0.078 | 0.029 ±0.007 | 0.107 ±0.021 | 0.119 ±0.008 | 0.213 ±0.027 |
| | $\eta_v$=0.3, $\tau_c$=4 (w/o pw) | 1.13 ±0.059 | 0.622 ±0.073 | 0.019 ±0.003 | 0.075 ±0.015 | 0.126 ±0.016 | 0.307 ±0.062 |
| Graph-UNets | $h$=5,$n$=9,$\eta_v$=0.3,$\tau_c$=4 | 1.06 ±0.171 | 0.859 ±0.092 | 0.041 ±0.007 | 0.057 ±0.010 | 0.153 ±0.012 | 0.34 5±0.133 |

