# OpenReview forum: "MeGraph: Capturing Long-Range Interactions by Alternating Local and Hierarchical Aggregation on Multi-Scaled Graph Hierarchy"
_NeurIPS.cc/2023/Conference — NeurIPS 2023 poster_

### Official Review · Reviewer_qUTa · 2023-07-02

**Soundness:** 2 fair
**Presentation:** 2 fair
**Contribution:** 2 fair
**Rating:** 6
**Confidence:** 4

**Summary:**

The authors propose a novel methodology for capturing long-range interactions in graph-based learning models, that is based on mult-scale graph construction and merging. Specifically, following the Select-Reduce-Connect framework [1], the authors introduce a novel pooling method called Edgepool  that enables refinement of the original graph structure in an arbitrary continuum. They then perform graph pooling in order to create multi-scale graph instances of the original graph, and merge the inter-graph and intra-graph connections through the proposed MeGraph framework.

The authors perform extensive experimentation to over 10 datasets. They, also, show a limited (compared with the number of trainable parameters) ablation study on four synthetic datasets. They, also, propose a rather novel benchmark, called Graph Theory Benchmark (testing properties: graph diameter, single source shortest path, graph eccentricity, maximum connected component), where they evaluate the prediction capabilities of their model. The results suggest a solid performance of MeGraph framework for tasks that require both shallow and deeper architectures.

[1] Daniele Grattarola, Daniele Zambon, Filippo Maria Bianchi, and Cesare Alippi. Understanding pooling in graph neural networks. IEEE
 Transactions on Neural Networks and Learning Systems, 2022.

**Strengths:**

1. The proposed framework is fairly novel. It shows some dependencies on previous frameworks (i.e. SRC framework needed for graph pooling).
2. The idea and methodology of the hierarchical pooling and merging, although it's not new (e.g. DiffPool), its quite interesting and intuitive for the problem of capturing long-range interactions.
3. The extensive experimentations to a broad set of datasets suggest a strong performance of the MeGraph framework.
4. There is a newly introduced benchmark, called Graph Theory Benchmark that can be used for various methods of evaluation.
5. The written text is fairly clearly written. However, it can be improved, mainly due to the fact that it's hard to follow the main contribution section. A more clear summarization of the contributions (methodology, and experimentation) would significantly improve the clarity of the paper.
6. The problem that MeGraph is approaching is quite important, mainly due to the known incapabilities of various graph neural network architectures to perform on par in deeper settings.

**Weaknesses:**

1. I am concerned about the computational complexity of the model. It seems that in the average case the combined computational cost of pooling, merging, and message passing is high, with respect to simple graph neural network models. The authors propose some variants to alleviate the high computational complexity, however it becomes more unclear how each one of the choices impact the behavior of the model. It seems (based on Tables 2, 4 ) that either the best performance comes from the computationally heavy variant of MeGraph or that there is no clear impact of the model choices.
2. The model consists of a high number of trainable parameters, and module choices. Although I value the effort of the authors for an ablation study, I think that a further investigation of the parameters of the modules ( especially with respect to the encoder-decoder choices, and the graph pooling methods) would be quite helpful.
3. There is no safe conclusion that long-range interactions can be captured because of the multi-scale method that the authors propose rather than the increased parameter space in MeGraph. The lack of any theoretical insight on why the specific choices contribute to capturing long-range interactions hinders convincing the reader that the multi-scale approach is appropriate for LRI.

**Questions:**

1. Can the authors pose any theoretical motivations on why the combination of SRC framework with the MeGraph framework can contribute to more effectively capture long-range interactions?



**Limitations:**

1. As I mentioned, I think that there is a lack of theoretical motivation for the choice of the model's modules. It would be quite helpful to provide any information on positive benefits of the SRC framework,as well as the representations obtained from the intra-graph and inter-graph interactions.
2. The computational complexity is dependent on the choice of pooling methods, and encoder-decoder methods (as well as the message passing steps). It would be quite interesting to investigate on any potential improvement that would balance the increased parameter space with the training time.
3. The written text can be further be clarified. The complex architecture of MeGraph requires a clear, and easy-to-follow description, so that the readers can grasp its mechanics.

---

> ### Author Rebuttal · Authors · 2023-08-09
>
> Thanks for your invaluable reviews. We provide point-by-point responses below.
>
> >0. The authors propose some variants to alleviate the high computational complexity, however it becomes more unclear how each one of the choices impact the behavior of the model. It seems ...
>
> We'd like to clarify that, when discussing methods to reduce time complexity in L215, we are referring to the variants described in Appendix C.5, not those in the baseline section (h=1 and n=1). The variants from Appendix C.5 primarily replace certain GN blocks with the Identity function to decrease time complexity. These include U-Shaped, Bridge-Shaped, and Staircase-Shaped structures and we present their running speed on the treecycle dataset in Table R7. To prevent confusion, we'll make necessary revisions.
>
> The h=1 and n=1 variants serve as baselines (refer to L224-226). Comparing the MeGraph model to its h=1 variant underscores the significance of using hierarchical (multi-scale) structures. When comparing the MeGraph model to its n=1 variant, it demonstrates the value of repeated information exchange throughout the mega graph hierarchy. We selected these two variants as baselines because it allows for fair comparisons by ensuring identical hyper-parameters, as detailed in Appendix F.1.
>
> >1. I am concerned about the computational complexity of the model. It seems that in the average case the combined computational cost of pooling, merging, and message passing is high, with respect to simple graph neural network models.
>
> In Section 3.4 (L213), we demonstrated the theoretical complexity of the MeGraph framework which is $1/(1−\eta)$ times that of a standard GNN. Here, $\eta$ is the ratio of the graph size before and after pooling. Notably, when $\eta \le 0.5$, $1/(1−\eta) \le 2$, suggesting the overall complexity is at most double that of the GNN when the graph size is halves for each pooling.
>
> However, in practice, the pooling method often targets the node ratio, but the edge ratio might be higher. To illustrate, we present the node and edge counts in a graph hierarchy from the synthetic treecycle dataset, using a randomized graph pooling method with a $\eta_v=0.3$ node pooling ratio:
>
> *Table R6*: Node and edge counts across hierarchies for a sample graph in the treecycle dataset.
> |count|h=1|h=2|h=3|
> |-|-|-|-|
> |intra-nodes|871|261|78|
> |intra-edges|972|964|850|
> |inter-nodes|1132|339|/|
> |inter-edges|1742|522|/|
>
> The edge count decreases at a slower rate compared to the node count. Furthermore, the overhead of inter-conv becomes nonnegligible when the edge and node counts are similar. We present the running times for edgepool and random-pool for heights 1 to 3 when $n=3$:
>
> *Table R7*: Training time (s) per epoch for various methods with n=3 layers on the treecycle dataset.
> |method|h=1|h=2|h=3|h=3 X-Pool|h=3 bridge|h=3 staircase|
> |-|-|-|-|-|-|-|
> |edgepool|0.026|0.068|0.127|0.078|0.098|0.081|
> |random  |0.026|0.056|0.111|0.064|0.087|0.070|
>
> To speed up, we can employ the X-Pool variant (refer L167) which approximately doubles the speed, but at a potential performance cost, as shown in the ablation study on cross update functions (L317). As discussed in Appendix C.5, substituting some GN blocks with identity functions can reduce runtime. The running times of two such variants (bridge and staircase) are displayed above, offering noticeable speed improvements. We leave exploring the balance between speed and performance in future studies.
>
> >2.a They show a limited ablation study on four synthetic datasets.
>
> We'd like to clarify that our ablation study is not limited to four synthetic datasets ([L280] Hierarchy vs. Locality), but also covers the Graph Theory Benchmark (for [L300] Varying S-EdgePool, [L314] Changing cross update function, [L319] Disabling bidirectional pathway) and real-world datasets (for [L310] Varying GN Block).
>
> >2.b ..., I think that a further investigation of the parameters of the modules ( especially with respect to the encoder-decoder choices, and the graph pooling methods) would be quite helpful.
>
> Thanks for your suggestions. As clarified above, we have conducted ablation studies on the graph pooling methods. We adjusted the hyper-parameters $\tau_c$ and $\eta_v$ for S-EdgePool and displayed MeGraph's performance with various poolings in Table 1. As stated in L309, irrespective of the pooling variant used, MeGraph consistently outperforms h = 1 baselines, suggesting the robustness of MeGraph against different structures of the mega graph.
>
> We implemented Louvain pooling [B0] and a random pooling that assigns nodes to a random cluster. The ablation study results for these new graph pooling methods are shown in the **Overall Response**.
>
> For shared components and hyper-parameters in standard GNNs and MeGraph, including encoders and decoders, their choices could influence model performance and the optimal choices are typically task-specific. As detailed in Appendix F.1 (Experimental Protocol), we first optimized the hyper-parameters for GFuN and selected the best configurations for each task. The effects of this tuning are seen in the performance differences between MeGraph(h=1) and GFuN, as shown in Tables 3, 4, 9 and 10-12. We will update Tables 10-12 accordingly.
>
> >3. Can the authors pose any theoretical motivations on why the combination of SRC framework with the MeGraph framework can contribute to more effectively capture long-range interactions?
>
> As discussed in Appendix D.1 (L714), hierarchical architecture like MeGraph provides a much smaller number of aggregation steps ($O(\log(|V|))$) for capturing long-range interactions than standard message-passing GNNs ($O(|V|)$).
>
> >4. A more clear summarization of the contributions would significantly improve the clarity of the paper. ... The complex architecture of MeGraph requires a clear, and easy-to-follow description, so that the readers can grasp its mechanics.
>
> Thanks for your suggestion. We will revise accordingly to make the text clearer.

---

### Official Review · Reviewer_Y5z6 · 2023-07-08

**Soundness:** 3 good
**Presentation:** 3 good
**Contribution:** 3 good
**Rating:** 5
**Confidence:** 5

**Summary:**

In summary, the paper proposes a mechanism to learn on graph in a multiscale and hierarchical manner. This is one of the approaches that can address the long-range problem, in which the graph has a large diameter, i.e. the length of maximum shortest paths between all pairs of nodes is long, in graph learning.

Given the following strengths and weaknesses, I vote this paper in the borderline. There are certainly more works to do.

**Strengths:**

I believe this paper asks the right question: Long-range problem in graph learning is certainly important. Existing GNNs based on the conventional message passing have severe limitations with long-range graphs (i.e. graphs with "long" maximum shortest paths). The reason is message passing only allows exchanging of information among neighboring nodes, and consequently we would need many layers / iterations of message passing (one can argue that the number of layers needed is proportional to the size of the graph) so that two distant nodes can reach each other. However, it is computationally infeasible to have many layers for large graphs and it will cause the problem of over-smoothing and potentially other problems with gradients and training.

Therefore, we need a method that allows distant nodes can exchange information via a small number of hops. Constructing a hierarchical and multiscale structure is one of the reasonable choices.

**Weaknesses:**

* Novelty: The long-range problem has attracted increasing attention from the graph learning community. There are similar ideas / works about multiscale and hierarchical structure learning being developed in parallel with this work.

For example, this work "Multiresolution equivariant graph variational autoencoder" https://iopscience.iop.org/article/10.1088/2632-2153/acc0d8/pdf proposed a similar multiscale / multiresolution method, but the construction of the hierarchy is done via a flexible learning-to-cluster algorithm in data-driven manner. The follow-up paper "Multiresolution Graph Transformers and Wavelet Positional Encoding for Learning Long-Range and Hierarchical Structures" https://arxiv.org/pdf/2302.08647.pdf (accepted at Journal of Chemical Physics) proposed an extension with Graph Transformers and a new positional encoding motivated from multiresolution analysis and wavelet theory, that can efficiently learn long-range and hierarchical graphs.

Furthermore, a recent ICML 2023 paper "On the Connection Between MPNN and Graph Transformer" https://arxiv.org/pdf/2301.11956.pdf suggested that a simple method of adding a virtual node (i.e. a node connecting to all other nodes) can significantly boost the performance of the conventional message passing for the long-range graphs. This work has theoretical analysis that proves the equivalence between adding a virtual node and graph transformer.

* Theoretical analysis: The paper got rejected from ICLR 2023 https://openreview.net/forum?id=Oz0npxjLAsI. One of the main reasons is the lack of theoretical analysis in analyzing the expressive power of their proposal. I suggest the authors to investigate theoretically their proposed model.

**Questions:**

N / A

**Limitations:**

I have written about other limitations of this work in the "Weaknesses" section.

I think the authors should extend the Graph Theory Benchmark further. For example, the graphs should be much larger, it is interesting to see how each method behaves on a very large and very long-range graphs in terms of efficiency and efficacy. In particular, since this work is about multiscale representation of a graph, the benchmark should have Stochastic Block Model (SBM) or some graphs with clustering or hierarchical patterns.

If the benchmark is sophisticated enough, the authors can submit to the data & benchmark track. Recently, there is a separate track for data & benchmark in main ML conferences.

---

> ### Author Rebuttal · Authors · 2023-08-09
>
> Thanks for your invaluable reviews. We provide point-by-point responses below.
> >1. Novelty: The long-range problem has attracted increasing attention from the graph learning community. There are similar ideas / works about multiscale and hierarchical structure learning being developed in parallel with this work.
>
> Thanks for pointing out these related works. Indeed, our work understands graphs beyond multiscale and hierarchical structure learning, and the key idea of our work and that of these works differ.
> * As stated in the abstract, the key idea of our work is to integrate the local and hierarchical structures of a multi-scale graph hierarchy into a single mega graph, and we proposed a MeGraph model that consists of multiple layers **alternating between local and hierarchical information aggregation on the mega graph**.
> * According to [B1], "The key idea of MGVAE is learning to construct a series of coarsened graphs along with a hierarchy of latent distributions in the encoding process while learning to decode each latent into the corresponding coarsened graph at every resolution level". MeGraph is different from MGVAE as it explicitly builds a mega graph upon the graph hierarchy and alternates local and hierarchical information aggregation on that, which is not seen in MGVAE. The targets are also different where MGVAE aims to generate graphs and MeGraph aims to capture both local and long-range interactions.
> * MGT [B2] first obtains a graph representation using GPS [42], builds substructures using the same clustering method as in [B1], and then applies Transformers on the substructures. The original graph and the substructures form a hierarchy of 2 layers. This method, based on a graph transformer, is a different way of capturing long-range interactions compared to ours. The WavePE [B2] also contains multiresolution information but is from the perspective of positional encoding.
> * MPNN+VN [B3] mainly focuses on proving the equivalence between adding a virtual node and graph transformer. Adding a virtual (root) node can be regarded as a very special case of our MeGraph architecture when the graph pooling module pools all nodes into a single node.
>
> Moreover, both MGT+WavePE [B2] and MPNN+VN [B3] conducted experiments in the Peptides-func dataset, while their results are inferior compared to ours (MGT+WavePE is 68.17% [B2], GatedGCN+RWSE+VN is 66.85% [B3], where MeGraph is 69.45%).
>
> We would include the discussions and their experimental results in our revision.
>
> [B1] Hy, T. S., & Kondor, R. (2023). Multiresolution equivariant graph variational autoencoder. Machine Learning: Science and Technology, 4(1), 015031.
>
> [B2] Ngo, N. K., Hy, T. S., & Kondor, R. (2023). Multiresolution graph transformers and wavelet positional encoding for learning long-range and hierarchical structures. The Journal of Chemical Physics, 159(3).
>
> [B3] Cai, C., Hy, T. S., Yu, R., & Wang, Y. (2023). On the connection between mpnn and graph transformer. arXiv preprint arXiv:2301.11956.
>
> >2. Theoretical analysis: The paper got rejected from ICLR 2023. One of the main reasons is the lack of theoretical analysis in analyzing the expressive power of their proposal. I suggest the authors to investigate theoretically their proposed model.
>
> We had provided supplementary theoretical explanations for the expressive power of MeGraph during that response period, and the reviewer appreciated our previous efforts. The according analysis is also provided in Appendix D.2 in the current submission. This explanation is also empirically supported by the results of using random poolings, which are provided in the Graph Poolings Ablation part of the **Overall Response**.
>
> In Appendix D.1, we have also rephrased the analysis provided in [43], which shows hierarchical architectures like MeGraph require a much smaller number of aggregation steps ($O(\log(|V|))$) for capturing long-range interactions than standard message-passing GNNs ($O(|V|)$).
>
> [43] Ladislav Rampášek and Guy Wolf. Hierarchical graph neural nets can capture long-range interactions. In 2021 IEEE 31st International Workshop on Machine Learning for Signal Processing (MLSP), pages 1–6. IEEE, 2021.
>
> >3. I think the authors should extend the Graph Theory Benchmark further. For example, the graphs should be much larger, it is interesting to see how each method behaves on a very large and very long-range graphs in terms of efficiency and efficacy. In particular, since this work is about multiscale representation of a graph, the benchmark should have Stochastic Block Model (SBM) or some graphs with clustering or hierarchical patterns.
>
> Thanks for your suggestion. We have created another version of our Graph Theory Benchmark. For each graph generation method, the generated dataset contains 500 graphs, and each graph contains 100 to 200 nodes. We also add a new graph generation method Stochastic Block Model (SBM). The generation details and the results are shown in the **Overall Response**.

---

### Official Review · Reviewer_NF28 · 2023-07-17

**Soundness:** 4 excellent
**Presentation:** 4 excellent
**Contribution:** 4 excellent
**Rating:** 8
**Confidence:** 3

**Summary:**

This paper proposes MeGraph, a GNN architecture that interleaves local and hierarchical structural information in a graph at multiple-scales, to capture long-range interactions (LRI). The authors propose S-EdgePool that generalizes EdgePool by allowing more than two nodes to be clustered in order to achieve a desired pooling ratio. Using graph pooling, they build a "mega graph", which includes inter-edges connecting nodes at one height to the corresponding super-nodes at the next height. Furthermore, they propose a message passing scheme based on "MeeLayers", which first propagates information locally at each height, then propagates upwards and then backwards in the hierarchy. A complexity analysis is provided. Authors conduct an extremely extensive set of experiments on graph theory benchmarks, 10 standard LRI tasks plus three others proposed in the paper, LRGB, GNN benchmarks and OGB. Results show the superior performance of MeGraph in comparison with GCN, GIN, GAT, GatedGCN, Graph U-Net and a large variety of baselines specific to LRGB. An ablation study demonstrates gains from larger height and number of layers, as well as some other components of the method.

**Strengths:**

S1. The paper investigates a fundamental problem: how to capture longe-range interactions in a graph using GNNs without incurring over-smoothing or over-squashing issues.
S2. The proposed solution is clean, well-justified and introduces some novel components (e.g., S-EdgePool, MeeLayer).
S3. The proposed solution matches or exceeds the performance of reference methods in a large variety of diverse benchmarks.
S4. Excellent presentation; writing is very clear despite the sophistication of the method.
S5. Extremely extensive set of experiments (9 out of 16 pages in the appendix contain additional tables/plots).
S6. Code and details of the experimental setup allow great reproducibility.

**Weaknesses:**

W1. There is no discussion regarding the hierarchies discovered by MeGraph for specific datasets/tasks, which could shed some light into the way the method is working.
W2. HGNet also builds a multi-scale hierarchy via graph pooling and was shown to slightly outperform Graph U-Net, but is not included in the comparisons.

**Questions:**

Q1. While perturbation-based graph explainability frameworks can be applied with MeGraph, they would not provide immediate insights into how the hierarchy comes into play. Is there any graph explainability frameworks suitable for this task or are there fundamental challenges in deriving explanations for outputs predicted by MeGraph?
Q2. Was there a particular reason not to include a comparison with HGNet?
Q3. Does S-EdgePool tend to create groups of nodes with roughly the same size? If so, how would the performance of MeGraph be affected in graphs where clusters of different sizes arise naturally?

**Limitations:**

Yes, the authors have discussed some obvious limitations in Section 6.

---

> ### Author Rebuttal · Authors · 2023-08-10
>
> Thanks for your invaluable reviews. We provide point-by-point responses below.
> >1. [W1] There is no discussion regarding the hierarchies discovered by MeGraph for specific datasets/tasks, which could shed some light into the way the method is working. [Q3] Does S-EdgePool tend to create groups of nodes with roughly the same size? If so, how would the performance of MeGraph be affected in graphs where clusters of different sizes arise naturally?
> Thanks for your suggestion, we plotted the graph hierarchies discovered by MeGraph in the shortest path tasks of Graph Theory Benchmark, in Figure 2 of the attached pdf (in Overall Response).
>
> For the S-EdgePool with $\eta_v=0.3, \tau_c=4$ (The 1st, 3rd, and 5th hierarchy), the size of the clusters are roughly the same size (as the cluster size is restricted as at most 4), and the structure of the graph is well preserved after pooling. For the S-EdgePool with $\eta_v=0.3$ (no cluster size limit), the size of the resulting cluster could vary, depending on the structure of the graph. In dense graphs (e.g. generated by the 'geo' method, 4th hierarchy in Figure 2 of the attached pdf), the size of the resulting cluster could be very large while leaving some node forms a cluster alone. The former S-EdgePool leads to better performance as indicated in Table 1.
>
> MeGraph still performs well when clusters of different sizes arise naturally. We use Stochastic Block Model (SBM) to generate graphs with clusters of different sizes to illustrate.  In Table R2, MeGraph with S-EdgePool still shows comparable or better performance compared to the baselines. Moreover, as shown in Figure 3 of the attached pdf, S-EdgePool preferentially performs clustering in the clusters of the original graph. For reference, we also show the Louvain pooling result in the right part of Figure 3, which can be regarded as a natural clustering in the original graph.
>
> >2. [W2] HGNet also builds a multi-scale hierarchy via graph pooling and was shown to slightly outperform Graph U-Net, but is not included in the comparisons. [Q2] Was there a particular reason not to include a comparison with HGNet?
>
> Thanks for your question. We thought HGNet has a similar architecture as Graph U-Net and could have similar issues, and therefore we didn't compare ours with HGNet. As suggested, we have included the comparison with HGNet, and the results are provided in Table R3 in **Overall Response** and Tables R4 and R5 below. It can be seen that MeGraph achieves better or comparable results than both Graph U-Nets and HGNet, where MeGraph achieves the best performance in most tasks (the best one is bolded in the table).
>
> *Table R4*: GNN Benchmark results (Corresponding to Table 3)
> | dataset |        ZINC         |        AQSOL        |       MNIST        |      CIFAR10       |      PATTERN       |      CLUSTER       |
> | :------ | :-----------------: | :-----------------: | :----------------: | :----------------: | :----------------: | :----------------: |
> | MeGraph | **0.2597 ± 0.0053** | **1.0017 ± 0.0210** | **97.860 ± 0.098** | **69.925 ± 0.631** | **86.507 ± 0.067** | **68.603 ± 0.101** |
> | U-Net   |   0.3320 ± 0.0103   |   1.0629 ± 0.0182   |   97.130 ± 0.227   |   68.567 ± 0.339   |   86.257 ± 0.078   |   50.371 ± 0.243   |
> | HGNet   |   0.4743 ± 0.0032   |   1.1192 ± 0.0101   |   90.122 ± 1.012   |   60.122 ± 0.428   |   69.448 ± 0.021   |   35.514 ± 0.046   |
>
> *Table R5*: OGB-G results (Corresponding to Table 4)
> | dataset |      molhiv      |     molbace      |     molbbbp      |    molclintox    |     molsider     |
> | :------ | :--------------: | :--------------: | :--------------: | :--------------: | :--------------: |
> | MeGraph |   77.20 ± 0.88   |   78.52 ± 2.51   |   69.57 ± 2.33   | **92.04 ± 2.19** |   59.01 ± 1.45   |
> | U-Net   | **79.48 ± 1.06** | **81.09 ± 1.66** | **71.10 ± 0.52** |   91.67 ± 1.69   | **59.38 ± 0.63** |
> | HGNet   |   77.96 ± 1.10   |   72.49 ± 0.93   |   70.26 ± 1.79   |   85.90 ± 0.90   |   58.91 ± 0.98   |
>
> | dataset |     moltox21     |    moltoxcast    |      molesol      |    molfreesolv    |      mollipo      |
> | :------ | :--------------: | :--------------: | :---------------: | :---------------: | :---------------: |
> | MeGraph | **78.11 ± 0.47** | **67.67 ± 0.53** | **0.886 ± 0.024** |   1.876 ± 0.058   |   0.726 ± 0.006   |
> | U-Net   |   77.85 ± 0.81   |   66.49 ± 0.45   |   1.002 ± 0.036   |   1.885 ± 0.069   |   0.716 ± 0.014   |
> | HGNet   |   77.85 ± 0.12   |   65.93 ± 0.61   |   0.924 ± 0.020   | **1.870 ± 0.056** | **0.706 ± 0.014** |
>
> >3.[Q1] While perturbation-based graph explainability frameworks can be applied with MeGraph, they would not provide immediate insights into how the hierarchy comes into play. Is there any graph explainability frameworks suitable for this task or are there fundamental challenges in deriving explanations for outputs predicted by MeGraph?
>
> Thanks for your question. We think perturbation-based graph explainability frameworks like GNNExplainer [57] could still be used. As explained in Section 3.2 Mega Graph Message Passing, we can regard the Mee layer as performing message passing over the mega graph. Though the mega graph keeps changing during the training stage, it will be fixed during testing time. Therefore, we can treat the mega graph as a standard graph and use GNNExplainer to explain the outputs predicted by MeGraph. The explanation would be a subgraph of the mega graph, containing the hierarchy information.

---

> > ### Comment · Reviewer_NF28 · 2023-08-10
> >
> > I confirm that I have read the authors' response and reviewed the additional PDF provided as part of the rebuttal. I appreciate the authors' effort in including yet another baseline (HGNet) despite its similarity with the U-Net architecture. All of it reinforces my previous evaluation that this is a well-rounded submission.

---

> > > ### Author Response · Authors · 2023-08-11
> > >
> > > We truly appreciate your consideration on our response and additional results. Your invaluable suggestions have helped us a lot in improving our work.

---

### Author Rebuttal · Authors · 2023-08-10

# Overall Response
We thank all reviewers for the consistently positive feedback and invaluable reviews. We provide point-by-point responses below by commenting on each of your reviews.

We report the following new results as suggested by the reviewers.
* As suggested by Reviewer Y5z6, we created another version of the Graph Theory Benchmark with larger graphs, and also include a new random graph generation method Stochastic Block Model (SBM).
* As suggested by Reviewer qUTa, we performed an ablation study on new graph pooling methods.
* As suggested by Reviewer NF28, we visualized the graph hierarchy discovered by MeGraph in the Graph Theory Benchmark. We compared two versions of S-EdgePool and found that the one that leads to better performance also better preserves the graph structure after pooling, suggesting the potential correlation.

For clarity in the response, we use Lxx to refer to Line xx in our submitted manuscript, e.g. L35 means Line 35.

## Larger Graphs in Graph Theory Benchmark
We created another version of our Graph Theory Benchmark. For each graph generation method, the generated dataset contains 500 graphs and each graph contains 100 to 200 nodes. For clarity in the response, we refer this version to as 'large' and the original version as 'medium', respectively.

We also add a new graph generation method Stochastic Block Model (SBM). We randomly sample the size of each block to be random from (5, 15), and the probability of edge within the block to be random from (0.3, 0.5) and those for other edges to be random from (0.005, 0.01). To make all the tasks well-defined, we filtered out the unconnected graphs during the generation.

The results are shown in Tables R1-R3 below.
* As shown in Table R1, the MeGraph model significantly outperforms the h=1 version and the Graph-UNets. The conclusion still holds (in Table R3) after adding the SBM graph generation method.
* Table R2 shows the results on the SBM-generated graphs only, where the shortest path among nodes is usually short as the nodes in the same block are usually well connected. In such graphs, Graph-UNets failed to identify local paths which standard GNNs (h=1 setting) can identify. This result matches our argument in our Introduction ([L35] hierarchical information propagation cannot take over the role of local information aggregation). In contrast, MeGraph performs comparable to or better than standard GNNs (h=1 setting).

*Table R1*: Graph Theory Benchmark (large version, averaged over different graph generation methods, excluding SBM generation):
|Category|Model|SP<sub>sssd</sub>|MCC|Diameter|SP<sub>ss</sub>|ECC|
|-------|----|----------------|---|-------|-------------|----|
|Megraph|$h=1,n=5$|360.813|34.865|208.490|360.017|237.996|
|Megraph|$h=5,n=5,\eta_v=0.3,\tau_c=4$|26.280|14.619|21.100|16.622|48.626|
|U-Net|$h=5,n=9,\eta_v=0.3,\tau_c=4$|110.998|28.052|43.794|110.998|81.322|


*Table R2*: Graph Theory Benchmark (large version, SBM generation only)
|Category|Model|SP<sub>sssd</sub>|MCC|Diameter|SP<sub>ss</sub>|ECC|
|-------|-----|---------------|-----|-------|-------------|-----|
|Megraph|$h=1,n=5$|0.023|104.048|0.4486|0.173|0.7500|
|Megraph|$h=5,n=5,\eta_v=0.3,\tau_c=4$|0.059|47.816|0.4033|0.078|0.4919|
|U-Net|$h=5,n=9,\eta_v=0.3,\tau_c=4$|1.117|62.839|0.6390|1.722|0.6770|

*Table R3*: Graph Theory Benchmark (large version, averaged over different graph generation methods, including SBM generation)
|Category|Model|SP<sub>sssd</sub>|MCC|Diameter|SP<sub>ss</sub>|ECC|
|-------|-----|-----------------|-------|-------|-------------|-------|
|Megraph|$h=1,n=5$|328.014|39.4772|189.577|324.033|219.746|
|Megraph|$h=5,n=5,\eta_v=0.3,\tau_c=4$|**23.8963**|**16.8321**|**19.2185**|**14.9676**|**44.9234**|
|U-Net|$h=5,n=9,\eta_v=0.3,\tau_c=4$|101.009|30.3711|39.8708|100.070|75.1185|
|HGNet[43]|$h=5,n=9,\eta_v=0.3,\tau_c=4$|413.793|45.188|299.189|420.884|359.354|

## Graph Poolings Ablation
We implemented the Louvain pooling [B0] and a random pooling method that assigns nodes to a random cluster. We conduct an ablation study for these two graph pooling methods on the treecycle and treegrid dataset, the results are illustrated in Figure 1 of the attached pdf.

For the random pooling, the hierarchical information becomes useless, and the MeGraph still achieves accuracy that is similar to the standard GNNs (h=1 variant), which supports our discussion in Appendix D.2 (MeGraph can degenerate to standard GNNs).

For Louvain pooling on the treecycle dataset, performance is marginally improved compared to EdgePool. This might be because the fixed clustering from the Louvain algorithm is better suited for this task compared to EdgePool. Such observations indicate the MeGraph architecture's robustness across various reasonable pooling methods.

[B0] Fast unfolding of communities in large networks, Vincent D Blondel, Jean-Loup Guillaume, Renaud Lambiotte, Renaud Lefebvre, Journal of Statistical Mechanics: Theory and Experiment 2008(10), P10008 (12pp)

## Plot Graph Hierarchy
We plotted the graph hierarchies discovered by MeGraph in the shortest path tasks of Graph Theory Benchmark, in Figures 2 and 3 of the attached pdf. In Figure 2, the S-EdgePool with $\eta_v=0.3, \tau_c=4$ well preserves the structure of the graph after pooling, while the S-EdgePool with $\eta_v=0.3$ (no cluster size limit) sometimes pooled too many nodes together, breaking the graph structure. The former S-EdgePool leads to better performance as indicated in Table 1. We also plot the hierarchy for SBM-generated graphs in Figure 3, indicating that EdgePool can handle graphs that naturally contains clusters of different size.

---

### Decision · Program_Chairs · 2023-09-21

**Decision:**

Accept (poster)

**Comment:**

In graph representation learning, message passing allows exchanging of information among neighboring nodes in order to create node representations. Gathering the information from distant parts of a graph, however, may need many message passing layers. Unfortunately, if many such layers are applied on large graphs, the node embeddings tend to a steady-state convergence (over-smoothing), which, combined with backpropagation over many layers creates difficulties computing accurate gradients.

Therefore, there is a desire for a method that allows distant nodes to exchange information via a small number of layers in the neural network. One such research direction constructs hierarchical multiscale graphs that facilitate the communication between distant nodes in the graph via iterative graph pooling. The basic ideas of iterative graph pooling are well known.

This work proposes a new way to propagate the messages through these multiscale graphs. The reviewers believe this method, while considering a well-understood concept, is a worthy addition to the existing literature on this research direction.